# Descriptions of Three New Species of the Genus *Acerataspis* Uchida, 1934 (Hymenoptera, Ichneumonidae, Metopiinae), with an Illustrated Identification Key to Extant Species

**DOI:** 10.3390/insects14040389

**Published:** 2023-04-17

**Authors:** Jing-Xian Liu, Alexey Reshchikov, Hua-Yan Chen

**Affiliations:** 1Department of Entomology, College of Plant Protection, South China Agricultural University, Guangzhou 510642, China; 2Institute of Eastern Himalaya Biodiversity Research, Dali University, Dali 671003, China; 3Key Laboratory of Plant Resources Conservation and Sustainable Utilization, South China Botanical Garden, Chinese Academy of Sciences, Guangzhou 510650, China

**Keywords:** Darwin wasps, Ichneumonoidea, parasitoid, DNA barcodes, new species

## Abstract

**Simple Summary:**

*Acerataspis* Uchida, 1934 is a genus of ichneumonid parasitoids distributed in the Oriental and Palaearctic regions. To date, only seven extant species have been recorded. The host of the genus is still not known. This paper reviews all valid extant species of the genus based on both morphological and molecular analyses and describes three new species.

**Abstract:**

The Asian genus *Acerataspis* Uchida, 1934 is reviewed based on both morphology and DNA barcodes. Ten species are recognized in total, of which three species from Yunnan Province of China are described as new: *Acerataspis maliae* sp. nov., *A. seperata* sp. nov. and *A. similis* sp. nov. The male of *A. fukienensis* Chao, 1957 is described and illustrated for the first time. The genus is recorded from Thailand and Southeast Asia for the first time. An illustrated key to all known extant species is provided. With the supplement of DNA barcodes, a few diagnostic morphological characters are found useful in species identification.

## 1. Introduction

The genus *Acerataspis* Uchida 1934 is a small genus in the subfamily Metopiinae (Hymenoptera, Ichneumonidae), previously comprising seven extant species all distributed in Asia [1,2,3,4,5,6], and one fossil from the Late Eocene of Colorado, USA [7]. Most of the species (*A. cruralis* Chiu, 1962, *A. formosana* Cushman, 1937, *A. fukienensis* Chao, 1957, *A. fusiformis* Morley, 1913, *A. szechuanensis* Chao, 1962) are known from the Oriental part of China [3,4,5,8,9,10,11], and two species, *A. clavata* Uchida, 1934 and *A. sinensis* Michener, 1940, range from the Eastern Palaearctic to the Oriental region [4,12,13,14,15]. *Acerataspis fusiformis* was described from the Karen Hills in Eastern Myanmar [1] and was recently recorded from the Guangxi Province of China [11]. Townes mentioned a few undescribed specimens of this species deposited at the American Entomological Institute (AEI) from Indonesia and Philippines [2]. There are also a number of specimens of unidentified barcoded species originating from Chattogram, Bangladesh at the BOLD “(available at http://bins.boldsystems.org/index.php/Public_SearchTerms?query=Acerataspis[tax], accessed on 1 February 2023)” and Cameron Highlands, Malaysia [16].

To date, nothing is known about the biology of the genus. However, phylogenetic analysis of the subfamily with *Acerataspis* species included showed a close relationship with the genus *Metopius* Panzer, 1806 [16], which are known to be koinobiont, larval-pupal endoparasitoids of Lepidoptera, mainly Lasiocampidae, Erebidae, Notodontinae, Noctuidae, and Geometridae [6,17]. Members of both genera are coloured with black and yellow stripes, which probably indicates mimicry of aculeate wasps of the subfamily Eumeninae (Hymenoptera, Vespidae) [18].

In this study, we examined species of *Acerataspis* from Asia and reviewed it based on morphological and molecular analyses, with an illustrated key to species.

## 2. Materials and Methods

The specimens examined are deposited in the following institutions (curators in parentheses):

CAS: California Academy of Sciences, California, USA (Robert Zuparko).

DEI: Deutsches Entomologisches Institute, Eberswalde, Germany (Andreas Taeger).

HKU: Hong Kong University, Hong Kong, China (Benoit Guénard).

IOZ: Institute of Zoology, Chinese Academy of Science, Beijing, China (Kui-Yan Zhang).

MCSN: Museo Civico di Storia Naturale “Giacomo Doria”, Genoa, Italy (Maria Tavano).

QSBG: Queen Sirikit Botanic Garden, Chiang Mai, Thailand (Wichai Srisuka).

SCAU: South China Agricultural University, Guangzhou, China (Jing-Xian Liu).

TARI: Taiwan Agricultural Research Institute, Taichung, China (Chi-Feng Lee).

Specimens were newly sampled for this study in Northern Thailand using Malaise traps during the “Tea Fauna” project (https://teafauna.com) in an old secondary *Decteracarpus* forest with *Camellia sinensis* var. *assamica* (Masters) Kitamura in the understory [19]. Specimens from Hong Kong were sampled using Malaise traps during long-term monitoring by Christophe Barthélémy [20]. Specimens from other parts of China, such as Guangdong, Yunnan and Hainan provinces, were either collected by Malaise trap or sweep net by Li Ma, Shi-Xiao Luo and Hua-Yan Chen.

Specimens were examined using a Zeiss (Stemi-508) stereomicroscope (Carl Zeiss AG, Jena, Germany). Photos were taken at SCAU using a Keyence VHX-5000 (Keyence Corporation, Osaka, Japan) and Leica S8AP0 stereomicroscope (Leica Camera AG, Wetzlar, Germany) mounted with a digital camera. Photos of the type specimen of *A. formosana* were taken at the DEI using a Leica DFC 495 digital camera (Leica Camera AG, Wetzlar, Germany) and M205 C microscope (Leica Camera AG, Wetzlar, Germany). Photos of the type specimen of *A. cruralis* were taken at TARI using a LEICA MC 190 HD (Leica Camera AG, Wetzlar, Germany). Composite images with an extended depth of field were created from stacks of images using Combine ZP software and arranged and partly enhanced using Ulead PhotoImpact X3.

Morphological terminology follows that of Broad et al. 2018 [17]. Abbreviations used in the text are as follows: POL = the shortest distance between the posterior ocelli; OD = the longest diameter of a posterior ocellus; OOL = the shortest distance between a posterior ocellus and compound eye. T1–T7 refers to the metasomal tergites 1–7.

To supplement molecular species identification of *Acerataspis* for future work, the “DNA barcoding” region—mainly for the Arthropod mitochondrial cytochrome oxidase subunit 1 (*CO1*)—was amplified from species with fresh available specimens. Genomic DNA was extracted from one leg of each specimen using the TIANamp Micro DNA Kit (Tiangen Biotech, Beijing, China), following the manufacturer’s protocols. Polymerase chain reactions (PCRs) were performed using Tks Gflex™ DNA Polymerase (Takara) with the LCO1490/HCO2198 primer pair [21] and conducted in a T100™ Thermal Cycler (Bio-Rad). Thermocycling conditions were: an initial denaturing step at 94 °C for 5 min, followed by 35 cycles of 94 °C for 30 s, 50 °C for 30 s, 72 °C for 30 s and an additional extension at 72 °C for 5 min. Amplicons were directly sequenced in both directions with forward and reverse primers on an Applied Biosystems (ABI) 3730XL by Guangzhou Tianyi Huiyuan Gene Technology Co., Ltd. (Guangzhou, China). Chromatograms were assembled with Geneious 11.0.3. All sequences generated from this study are deposited in GenBank (for accession numbers see Table 1).

Sequences of *Acerataspis clavata* (Uchida, 1934) were downloaded from Genbank. Sequences were aligned by codons using MUSCLE implemented in Geneious. The p-distances within and between species were calculated in MEGA 11 [22]. The aligned sequences were then analyzed using RAxML as implemented in Geneious ver. 11.0.3 to generate a maximum likelihood (ML) tree to show the affinities among the studied species. Each codon position of the *CO1* sequence was treated as a different data partition, and the best-fit substitution model was determined using ModelFinder [23]. The best-fit model according to the Bayesian Information Criterion was GTR + Γ+I. Automatic bootstopping criterion was selected as the appropriate number of bootstraps; 300 replicates were run. A sequence of *Chorinaeus funebris* (Gravenhorst, 1829) (Ichneumonidae, Metopiinae) (KR931343) was downloaded from Genbank and used as an outgroup.

## 3. Results

### 3.1. Morphological Examination

In this study, 35 specimens representing ten morphospecies were examined from China, Myanmar, Thailand, and Russia, and ten morphospecies were recognized. Seven species were identified as previously described species and three are considered as new to science species as described below (Table 1 and Table 2). *Acerataspis clavata* and *A. fusiformis* are recorded from Hong Kong and Thailand; these are the first documented records of the genus from these regions as well as Southeast Asia.

### 3.2. DNA Barcoding

The present study generated 27 *CO1* sequences, representing six morphospecies with 642–684 bps. When sequences from GenBank and BOLD databases were compared using BLAST, only the sequences of *Acerataspis clavata* (Uchida, 1934) received close matches with two sequences also labeled as *A. clavata* (Uchida, 1934), but only share a 92.7% similarity. Genetic distances of the 27 sequences are provided in Appendix A. Intraspecific distances of the *CO1* sequences are generally less than 4%. Interspecific distances range between 9.7% and 16.5% (Appendix A). Each species recovered on the tree is clearly separated from all neighboring species, as shown in Figure 1.

### 3.3. Systematics

***Acerataspis* Uchida, 1934** (Figure 2 and Figure 3).

*Ceratapsis* Uchida, 1934, 275 [24]. Preoccupied by *Cerataspis* Gray, 1847. Type: *Cerataspis clavata* Uchida. Original designation.

*Acerataspis* Uchida, 1934, 23 [12]. New name for *Cerataspis* Uchida, not Gray. Type species: *Acerataspis clavata* Uchida, by original designation.

**Diagnosis.** Fore wing length 7.0–10.0 mm. Face and clypeus evenly moderately convex (Figure 2A). Clypeal sulcus absent. Upper margin of face continued dorsally as an intra-antennal projection with a deep central groove. Temple very short, mostly flat (Figure 2B). Occipital carina complete, but not reaching mandibular base ventrally, nor meeting hypostomal carina (Figure 2C). Subocular sulcus absent. Mandible with upper tooth a little larger than lower tooth. Notaulus absent. Scutellum short, transverse, prolonged lateroposteriorly as a tooth (Figure 2D). Propodeum with areas, anterior transverse carina present. Sternaulus broadly impressed. Metapleuron with juxtacoxal carina complete or incomplete. Fore wing with areolet sessile or petiolate, receiving 2m-cu vein before middle (Figure 2E), hind wing with nervellus interrupted below middle. Middle tibia with a single slender spur in male, with two spurs in female. Hind tibia with two spurs. All claws pectinate (Figure 2F). Metasoma clavate, T1–T3 with a pair of lateromedian longitudinal carinae from base to apex (Figure 3A), T4 of some species also with a pair of short lateromedian longitudinal carinae on anterior base, T6 rounded and turned under (Figure 3C,D). Ovipositor short, thin, tapered to apex, without a dorsal subapical notch (Figure 3B).

**Comment**. In previous studies of Metopiinae, the presence or absence of pectinae of tarsal claws were used to separate species in the genus *Hypsicera* [4,25]. However, this character was more or less overlooked or roughly described by authors in previous studies of *Acerataspis*. Based on the materials of *Acerataspis* from China, we found that the tarsal claw pectination of the female is a useful diagnostic character for species identification (although sometimes the pattern of pectination in the hind legs of some female specimens may vary between left hind leg and right hind leg). The proximal pecten of the claw is also very small and needs to be observed at high magnification.

**Biology.** No host records have been reported.

**Distribution.** Palaearctic and Oriental regions [6].

**Figure 2 insects-14-00389-f002:**
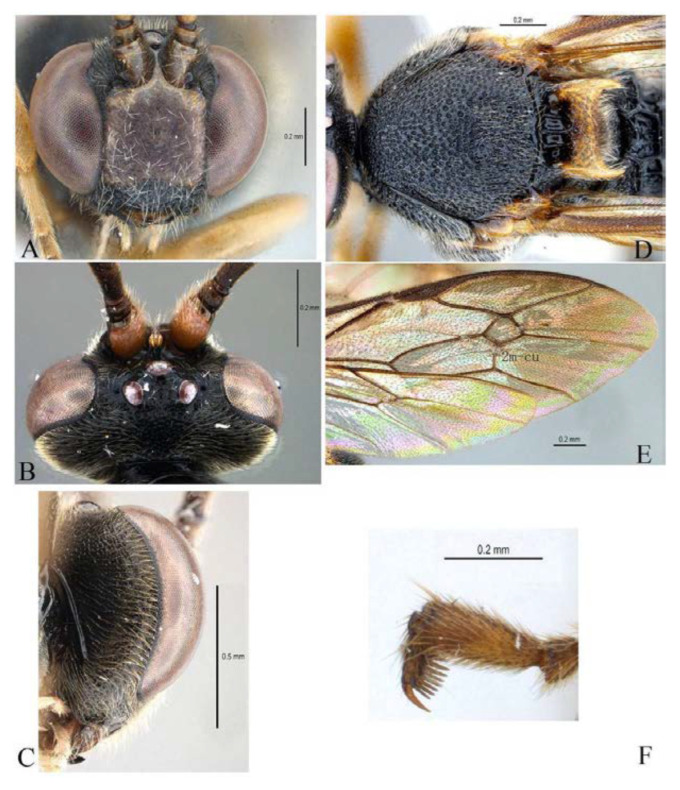
*Acerataspis* spp (*A. clavata*: (**A**,**D**–**F**); *A. fusiformis*: (**B**,**C**)). (**A**) Head, frontal view; (**B**) head, dorsal view; (**C**) head, posterior lateral view; (**D**) mesosoma, dorsal view; (**E**) fore wing; (**F**) fore tarsal claw of female.

**Figure 3 insects-14-00389-f003:**
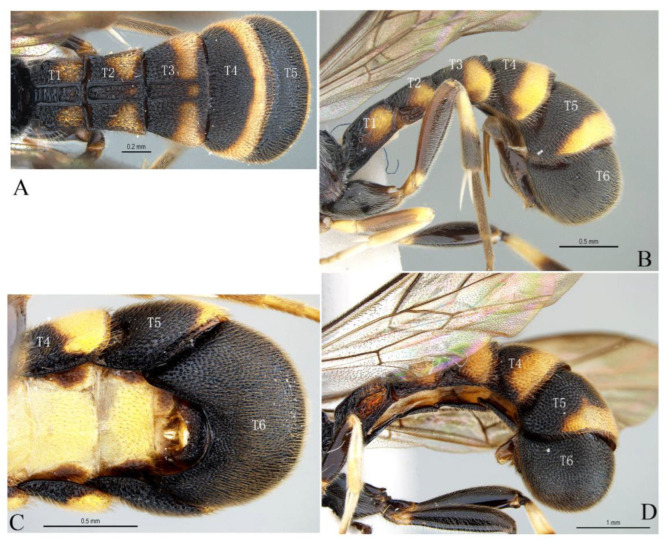
*Acerataspis* spp (*A. clavata*: (**A**,**B**,**D**); *A. fusiformis*: (**C**)). (**A**) Metasoma, dorsal view (female); (**B**) metasoma, lateral view (female); (**C**) T4–T6 of male specimen, central view; (**D**) metasoma of male, lateral view.


**Key to species of *Acerataspis* (only *A. formosa*, *A. clavata*, *A. fukienensis*, *A. fusiformis*, *A. separate* sp. nov. known from both sexes)**


1. Female (1a) …2

-. Male (1aa) …11



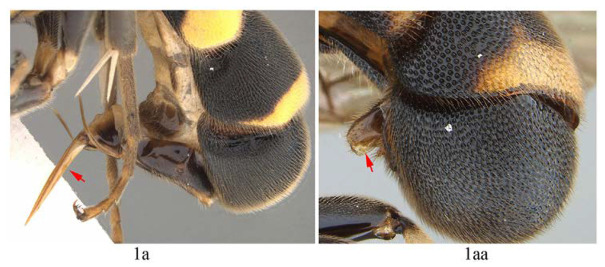



2. Hind tibia entirely black or blackish (2a) …3

-. Hind tibia with proximal half white or yellow (2aa) …6



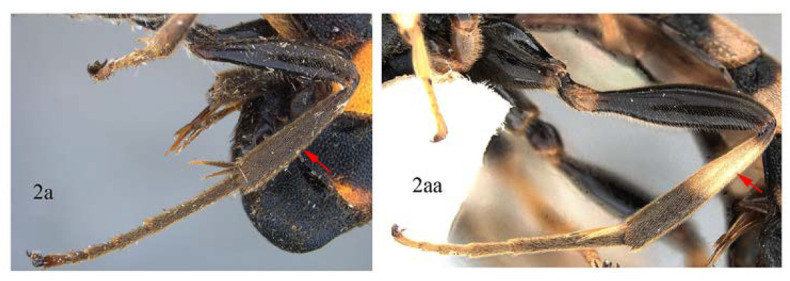



3. Fore wing with 1cu-a opposite M&RS (3a); fore tarsal claw with 3–4 large pectinae (3b); T4 with a pair of short lateromedian longitudinal carinae on anterior half (3c) …4

-. Fore wing with 1cu-a postfurcal to M&RS (3aa); claw with six or more fine pectinae (3bb); T4 without caraine (3cc) or a single of very short median longitudinal carinae on anterior half (3dd) …5



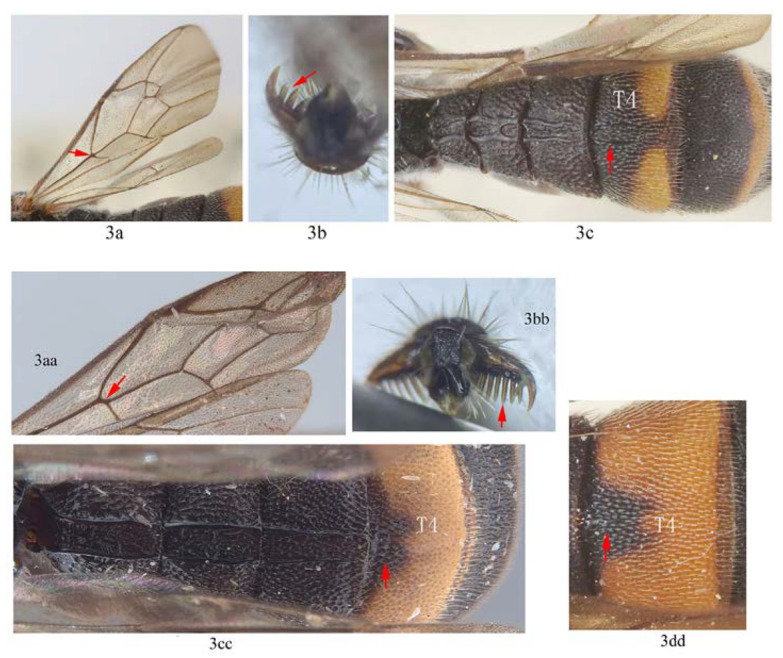



4. Fore wing with areolet sessile (4a); T1–T3 with yellowish or whitish marks (4b); T4 with apical yellow band complete (4b) …*A. formosana*

-. Fore wing with areolet petiolate (4aa); T1–T3 black (4bb); T4 with apical yellow band interrupted dorsomedially (4cc) …*A. fukienensis*



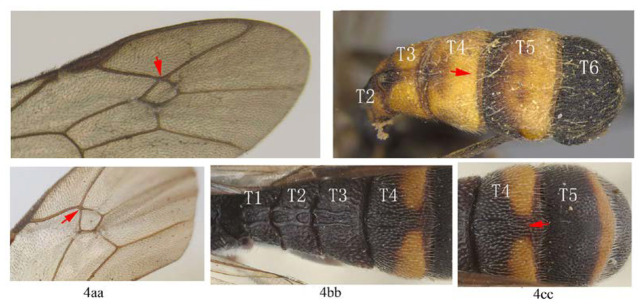



5. Scutellum entirely black (5a); subtegular ridge blackish brown (5b); T2 punctate and without irregular longitudinal carina between the latero-median longitudinal carinae (5c); T4 without carina anteriorly (5d) …*A. sinensis*

-. Scutellum anteriorly black and posteriorly largely orange yellow (5aa); subtegular ridge entirely orange yellow (5bb); T2 with an irregular longitudinal carina between the latero-median longitudinal carinae (5cc); T4 with a very short and weakly defined carina anteriorly medially (5dd) …*A. szechuanensis*



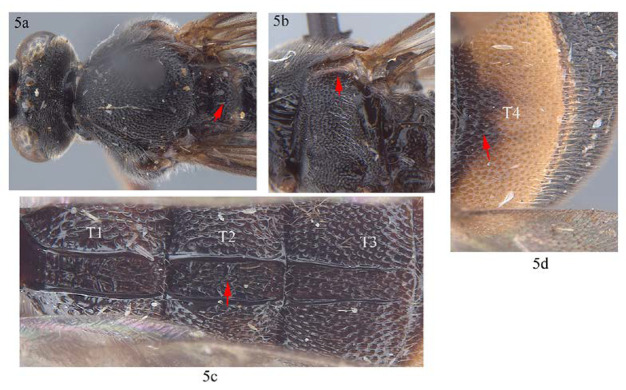





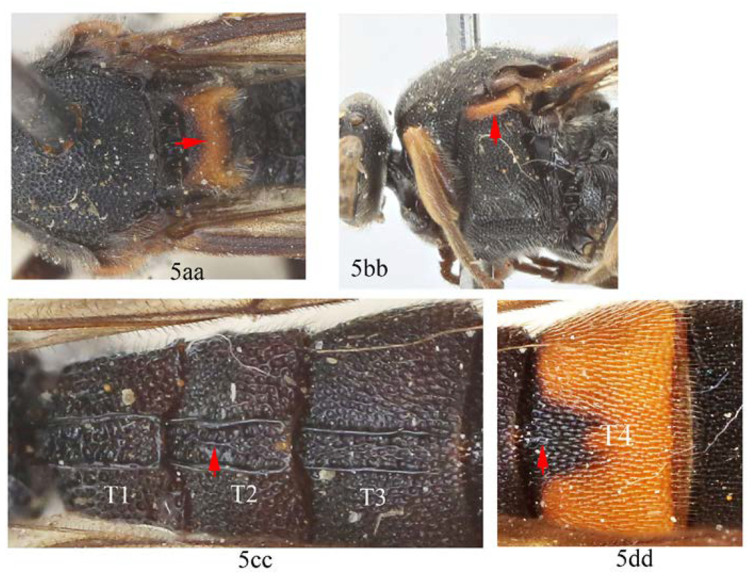



6. T4 with posterior yellow or whitish band dorso-medially interrupted (6a) …*A. separata* sp. nov.

-. T4 with posterior yellow or whitish band complete (6aa) …7



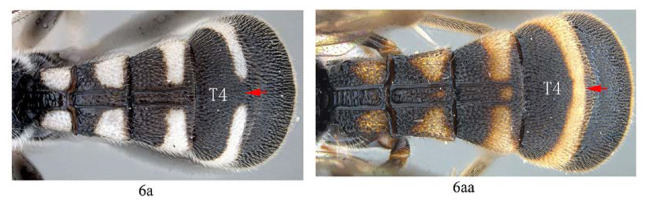



7. Fore tarsal claw with 3–4 pectinae (7a); fore wing with 1cu-a opposite M&RS (7b) (except for *Acerataspis maliae* sp.nov., 1cu-a slightly postfurcal to M&RS) …8

-. Fore tarsal claw with eight pectinae (7aa); fore wing with 1cu-a postfurcal to M&RS (7bb) …10



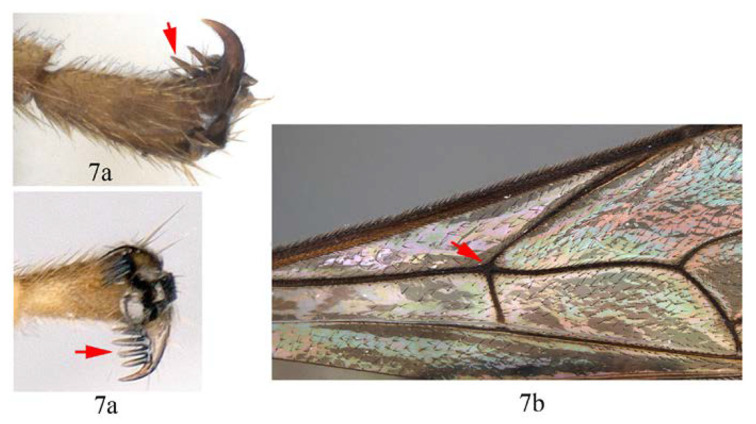





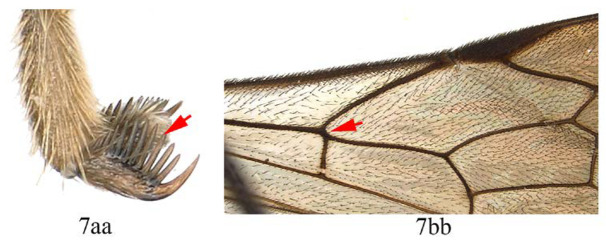



8. Hind tarsal claw with seven unequal pectinae (8a), fore wing with 1cu-a slightly postfurcal to M&RS (8b) …*A. maliae* sp. nov.

-. Hind tarsal claw with three or four unequal pectinae (8aa), fore wing with 1cu-a opposite M&RS (8bb) …9



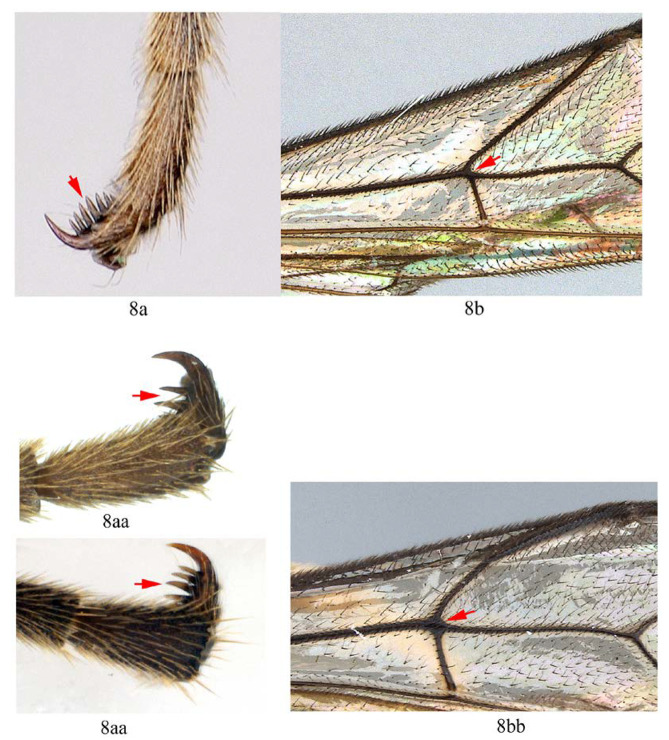



9. Hind tarsal claw with three pectinae (9a); metanotum orange yellow (9b) …*A. fusiformis*

-. Hind tarsal claw with four pectinae (9aa); metanotum black (9bb) …*A. similis* sp. nov.



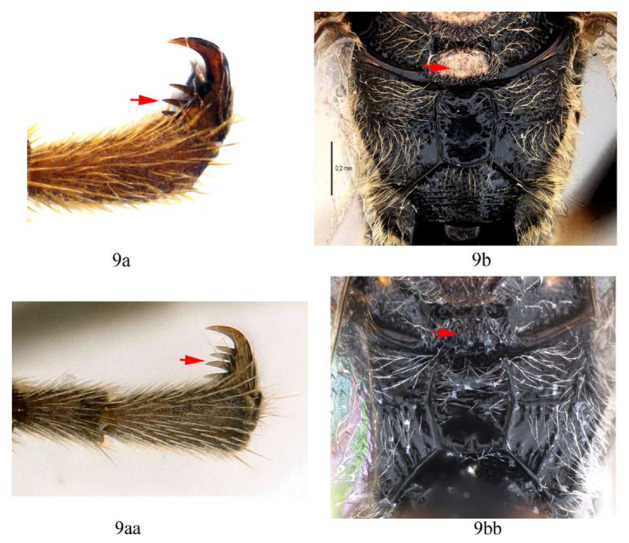



10. T2 and T3 entirely ferruginous, posterior margin of T2 straight (10a); T4 largely ferruginous with the extreme base black (10b) …*A. cruralis*

-. T2 and T3 anteriorly black and posteriorly yellowish brown, posterior margin of T2 convex (10aa); T4 black on anterior half and yellow on posterior half (10bb) …*A. clavata*



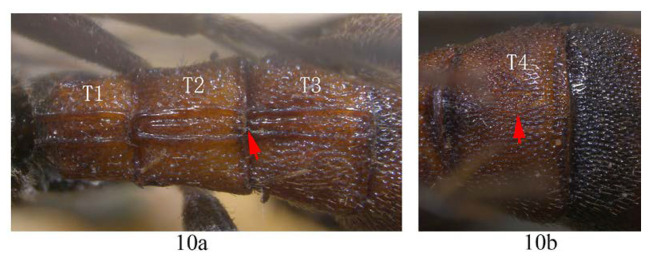





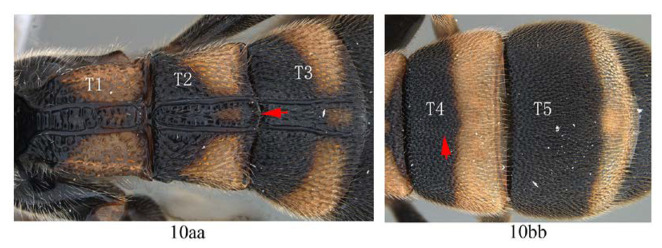



11. T4 with a pair of short lateromedian longitudinal carinae on anterior 0.45 of tergite (11a); hind tibia black (11b) …12

-. T4 without a pair of short lateromedian longitudinal carinae anteriorly (11aa); hind tibia proximally yellow or white (11bb) …13



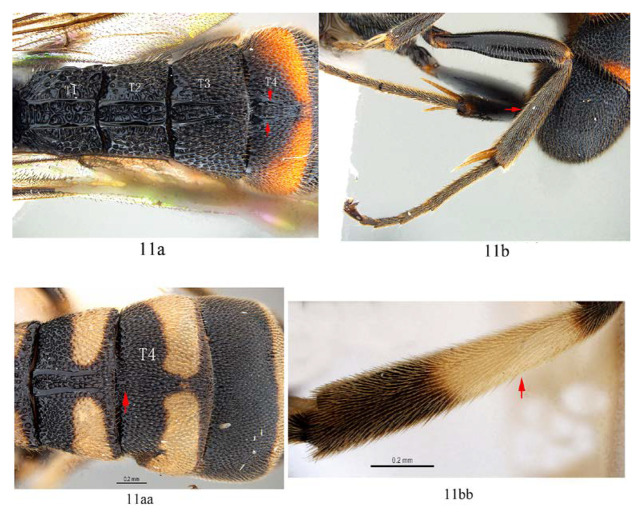



12. T1–T3 entirely black (12a) …*A. fukienensis*

-. T1–T3 with fuscous or brownish marks…*A. formosana*



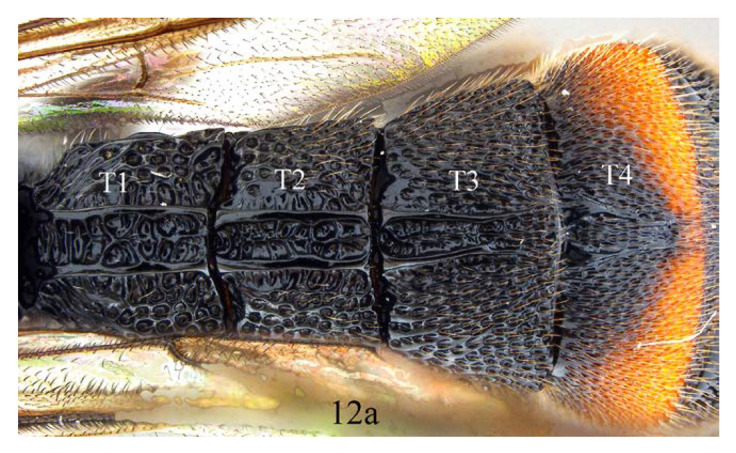



13. Fore wing with 1cu-a opposite M&RS (13a) …14

-. Fore wing with 1cu-a postfurcal to M&RS (13aa) …15



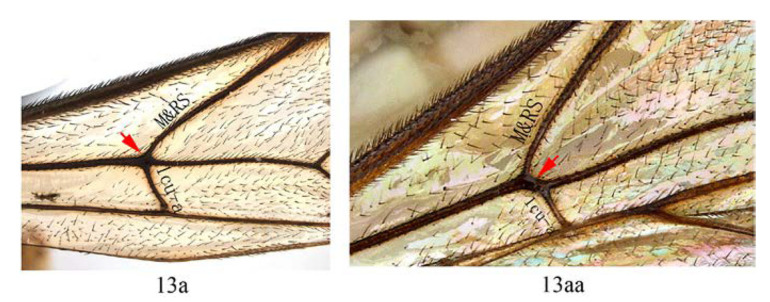



14. Hind tarsal claw with 3–4 pectinae (14a); metanotum yellow (14b); area superomedia of propodeum longer than wide (14c) …*A. fusiformis*

-. Hind tarsal claw with six pectinae (14aa); metanotum black (14bb); area superomedia of propodeum wider than long (14cc) …*A. clavata*



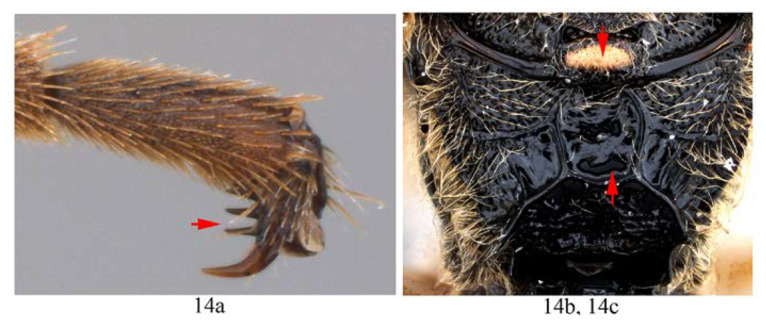





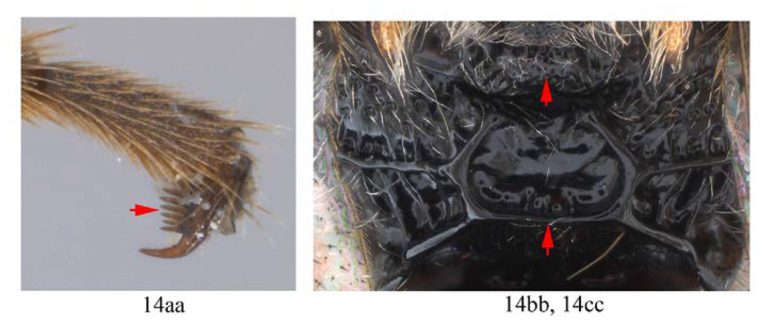



15. Hind tarsal claw with 3–4 pectinae (15a); metanotum yellow (15b) …*A. fusiformis*

-. Hind tarsal claw with more than five pectinae (15aa), metanotum black (15bb) …16



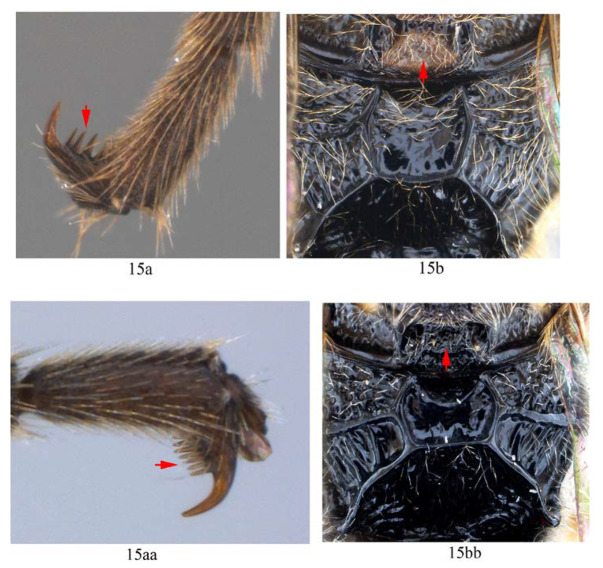



16. T4 with posterior white band interrupted dorso-medially(16a); terminal flagellomere stout and 2.8 times as long as wide (16b) …*A. separata* sp. nov.

-. T4 with posterior white band complete (16aa); terminal flagellomere slender and 5.0–5.4 times as long as wide (16bb) …*A. clavata*



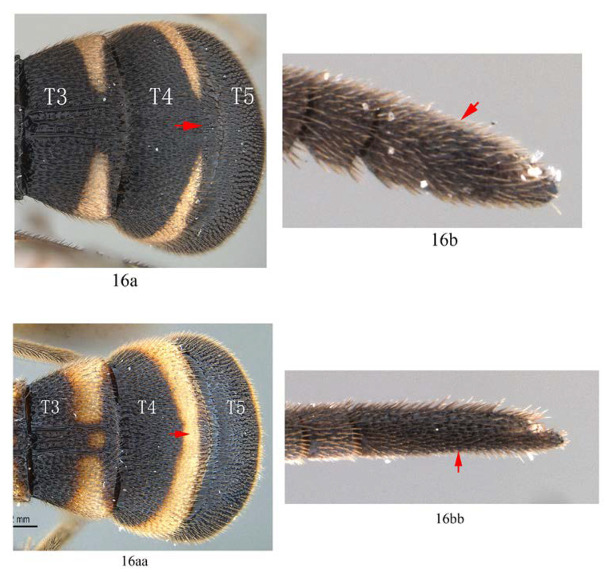



### 3.4. Species Descriptions

#### 3.4.1. *Acerataspis clavata* (Uchida, 1934) (Figure 4, Figure 5 and Figure 6)

*Cerataspis clavata* Uchida, 1934, Transactions of the Sapporo Natural History Society. 13(3): 275 [12]. Type, male, from Japan.

**Materials examined.** CHINA: 1♀1♂, Yunnan Province, Xishuangbanna Tropical Botanical Garden of the Chinese Academy of Science, 15.V–17.VI.2018, Malaise trap, 21°52′ N, 101°19′ E, alt. 543 m, Li Ma leg., SCAU-3011979, SCAU-3011983 (SCAU); 2♂, Yunnan, Xishuangbanna Botanical Garden, rubber plantation, 21°52′ N, 101°19′ E, alt. 543 m, Malaise trap, 16.III–11.IV.2021, Li Ma leg., SCBG-E0000053, SCBG-E0000055 (SCAU); 1♂, Yunnan Province, Xishuangbanna Tropical Botany Garden of Chinese Academy of Science, Rain Forest, 17.V–18.VI.2018, Malaise trap, alt. 560 m, 21°52′ N, 101°19′ E, Li Ma leg., SCAU-3011980 (SCAU); 1♂, Guangdong Province, Shenzhen, Tanglangshan, LSX-653, GD8, 30.VI–31.VIII.2020, 22°34′ N, 113°58′E, Luo Shi-xiao leg., Malaise trap, SCAU-3011049 (SCAU); 1♀, Guangdong Province, Shenzhen, Fairylake Botanical Garden, 1.VIII–8.IX.2020, 22°34′ N, 114°10′ E, Luo Shi-xiao leg., SCAU-3011050 (SCAU); 1♂, Hong Kong, Ping Shan Chai, 22°29′8.30″ N 114°11′14.74″ E, alt. 140 m, Malaise trap (M348), 14–30.IV.2018, Christophe Barthélémy leg., SCAU-3013714 (HKU); RUSSIA: 1♀, Primorje, Vladivostok, Sedanka, VII.1920, René Malaise leg., NHRS-HEVA000020635 (NHRS); THAILAND: 1♀, Chiang Mai, Mae Taeng, Pa Pae, 19°14′30.6″ N, 98°39′14.1″ E, old secondary forest with *Camellia sinensis* var. *assamica*, Malaise trap (Data#1). 21.III–11.IV.2017, Monsoon Tea leg. SCAU-3013718 (QSBG).

**Diagnosis.** Fore wing (Figure 4) with 1cu-a postfurcal to M&RS, areolet sessile. Propodeum with anterior transverse carina joining area superomedia at middle. T1–T3 with a pair of median carinae. All tarsal claws (Figure 6A–C) of female and male densely pectinate, with at least six pectinae on each claw. Antenna with scape largely black, only ventrally yellow. Hind tibia with proximal 0.4 yellowish white. T1–T3 with apical yellow band (band sometimes interrupted medially), T4 with apical margin yellow, usually not interrupted dorsomedially (Figure 4 and Figure 5A).

**Variation.** One specimen (specimen ID: SCAU-3013718) from Thailand with five pectinae on fore claw (Figure 6D) and apical yellow band of T4 dorsomedially interrupted (Figure 5B).

**Distribution.** Oriental and Eastern Palaearctic regions (newly recorded from Hong Kong and Thailand).

**Differential diagnosis.** This species is very similar to *A. fusiformis* (Morley, 1913) in colour pattern, it is slightly different from the latter in having all tarsal claws densely pectinate (female fore tarsal claw of *A. fusiformis* with only three pectinae), area superomedia of male laterally expanded (based on the materials from China, area superomedia weakly convergent posteriorly), fore wing with 1cu-a postfurcal to M&RS (fore wing with 1cu-a opposite M&RS of *A. fusiformis*), metanotum usually black (except one specimen from Yunnan slightly brown) (metascutellum distinctly yellow in *A. fusiformis*).

**Comments.** The specimen (ID: SCAU-3013718) from Thailand with five pectinae on fore tarsal claw and apical transverse band on T4 interrupted dorsomedially is an unusual variation within species, and it was determined based on DNA barcoding.

**Figure 4 insects-14-00389-f004:**
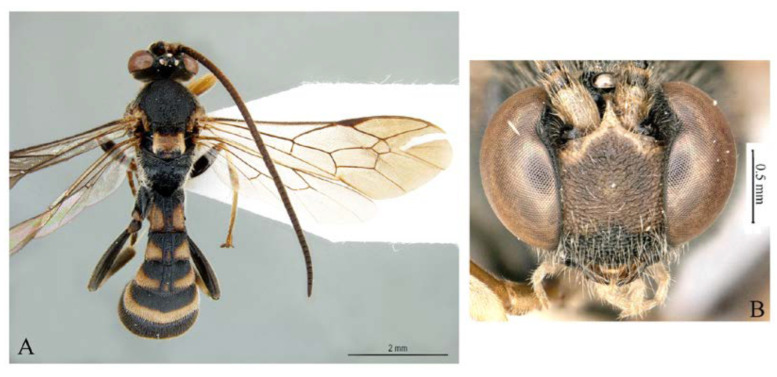
*Acerataspis clavata* (Uchida, 1934), female. (**A**) habitus, dorsal view; (**B**) head, frontal view (non-type specimen from China, specimen ID: SCAU-3011979). (Note: Facial colouration was altered after the DNA extraction experiment; the original colour of the species is yellow).

**Figure 5 insects-14-00389-f005:**
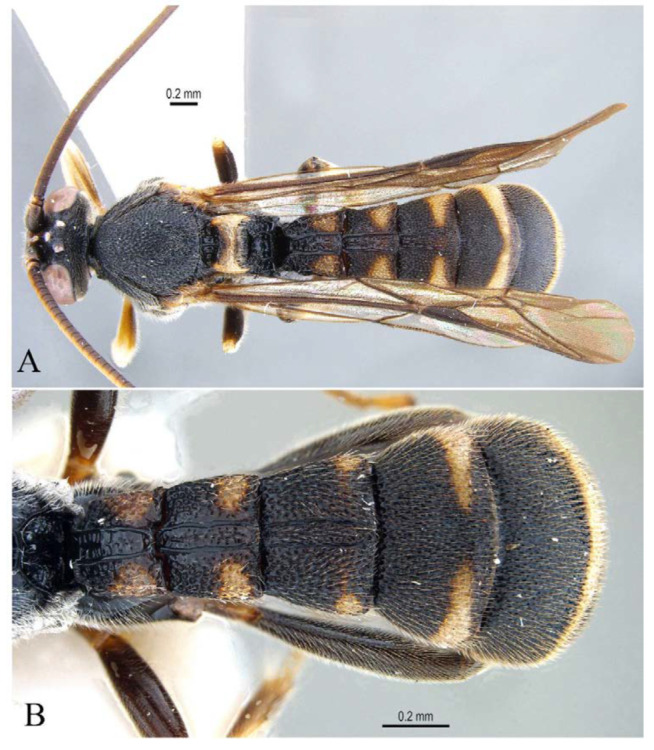
*Acerataspis clavata* (Uchida, 1934). (**A**) Male habitus, dorsal view (non-type specimen from China, specimen ID: SCAU-3011980); (**B**) metasoma of a female specimen from Thailand, dorsal view.

**Figure 6 insects-14-00389-f006:**
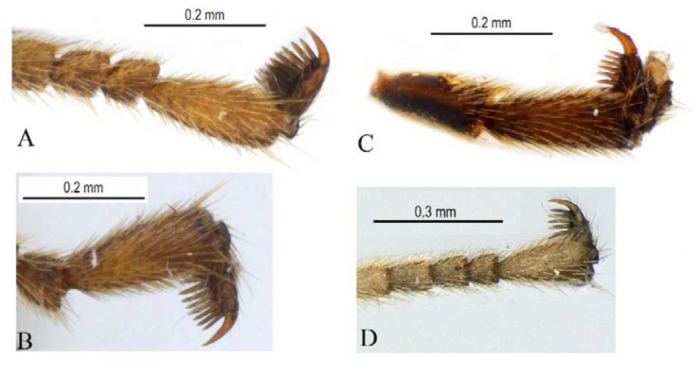
Tarsal claws of *Acerataspis clavata* (female). (**A**) Fore tarsal claw; (**B**) mid tarsal claw; (**C**) hind tarsal claw ((**A**–**C**), specimen ID: SCAU3011979, from China); (**D**) fore tarsal claw with five pectinae (specimen from Thailand).

#### 3.4.2. *Acerataspis cruralis* Chiu, 1962 (Figure 7 and Figure 8)

*Acerataspis cruralis* Chiu, 1962, 20: 5 [4]. Holotype, female, from Taiwan.

**Materials examined.** Holotype, 1♀, Labeled “Oiwake (Wushe), FORMOSA, 2. IX. 1940, Col. J. Sonan; Holotpye, type, S.C. Chiu 197; Holotype ♀, *Acerataspis cruralis*, DET.S.C. CHIU” (TARI).

**Diagnosis.** Propedeum with anterior transverse carina joining area superomedia at middle, area dentipara densely and finely punctate. Fore wing (Figure 8D) with 1cu-a postfurcal to M&RS; areolet sessile. T1 with a pair of carinae (Figure 8B); T2 with three longitudinal median carinae (Figure 8B), the median one obscure with apical half absent; T3 with three longitudinal carinae, the median one weak (Figure 8B); T4 strongly punctate, without carina (Figure 8E). Fore tarsal claw strongly pectinate, with 6–7 pectinae (Figure 8C). Hind leg black, proximal half of tibia yellow with proximal base black (Figure 7). Metanotum yellow. Metasoma ferruginous (Figure 7), with base of T1, base of T4, anterior half of T5, and T6 blackish to black.

**Distribution.** Oriental region [26].

**Differential diagnosis.** This species is very similar to *A. clavata*, especially in colour pattern, with differences between them as follows: *A. cruralis* with metascutellum yellow and T1 (except for the base black) to T4 (except for the extreme base black) mainly ferruginous, (*A. clavata* with metascutellum entirely black, T1–T4 black, apical margin with yellow marks or yellow band). Hind margin of T2 straight in *A. cruralis* (convex in its middle in *A. clavata*), fore claw weakly curved (Figure 8C) in *A. cruralis* (more strongly curved in *A. clavata*) (Figure 6A).

**Figure 7 insects-14-00389-f007:**
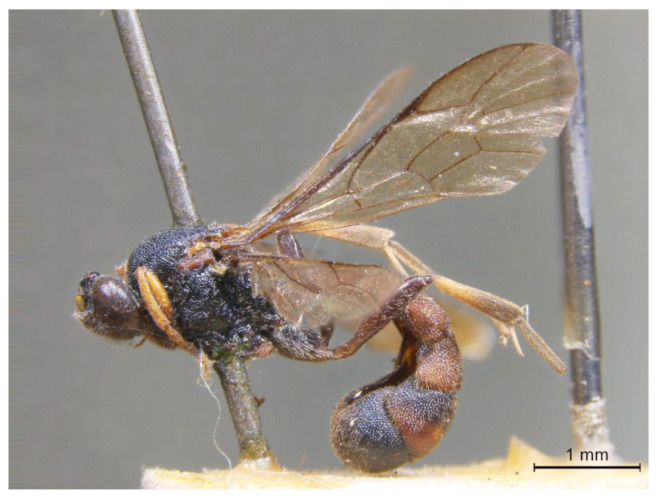
*Acerataspis cruralis* Chiu, habitus of holotype female.

**Figure 8 insects-14-00389-f008:**
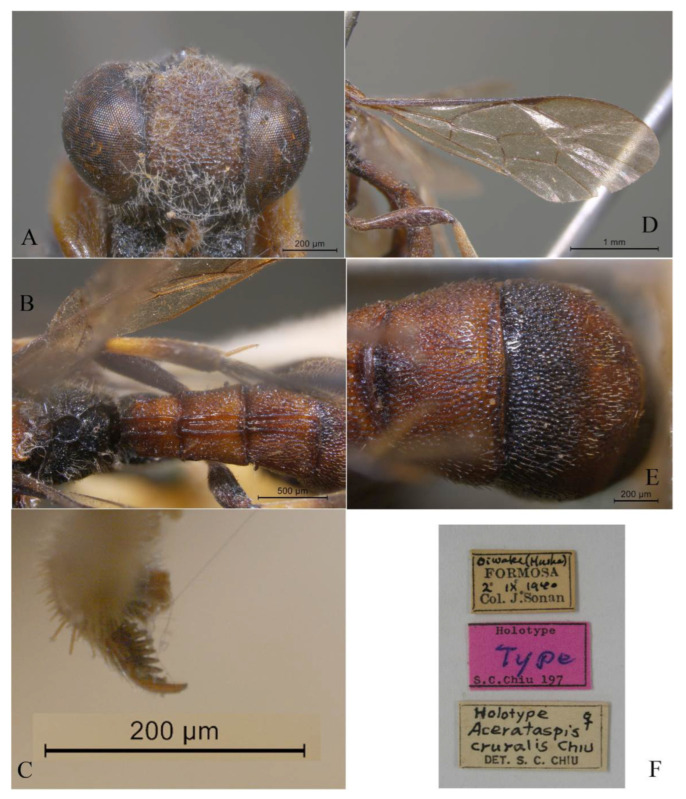
*Acerataspis cruralis* Chiu, holotype female. (**A**) Head, frontal view; (**B**) propodeum and T1–T3, dorsal view; (**C**) fore tarsal claw; (**D**) fore wing; (**E**) T4 and T5, dorsal view; (**F**) labels of holotype.

#### 3.4.3. *Acerataspis formosana* Cushman, 1937 (Figure 9 and Figure 10)

*Acerataspis formosana* Cushman, 1937, 4: 291 [8]. Type female.

**Material examined.** Holotype. 1♀, labelled “Suisharyo, Formosa, H.Sauter, X. 1911; Holotypus; *Acerataspis formosana*, Type, Cush; DEI-GISHym, 16611” (DEI). Figure posted on 21.iii.2019, 18:33, by Andreas Taeger. (Note: Type materials were examined based on the type photos provided by Andreas Taeger.)

**Diagnosis.** Propedeum with the area dentipara rugose. Fore tarsal claw (Figure 9H) with 2–3 large teeth. Fore wing with 1cu-a opposite to M&RS (Figure 10C). T1 and T2 with a pair of median longitudinal carinae (Figure 9D). T3 with median carinae distinct on anterior half, obscure and weakened from middle to apex; T4 anteriorly with a pair of short lateromedian longitudinal carinae (Figure 9C). Hind leg entirely black (Figure 10B,D).

**Differential diagnosis.** This species is similar to *A. clavata* in the colour pattern of metasoma, with differences between them as follows: *A. formosana* with hind tibia entirely black or blackish brown (hind tibia of *A. clavata* proximally with a white band), *A. formosana* with fore wing with 1cu-a vein opposite M&RS (fore wing with 1cu-a postfurcal to M&RS in *A. clavata*), T4 anteriorly with a pair of short lateromedian longitudinal carinae (Figure 9C) (T4 without a pair of short lateromedian longitudinal carinae in *A. clavata*).

**Figure 9 insects-14-00389-f009:**
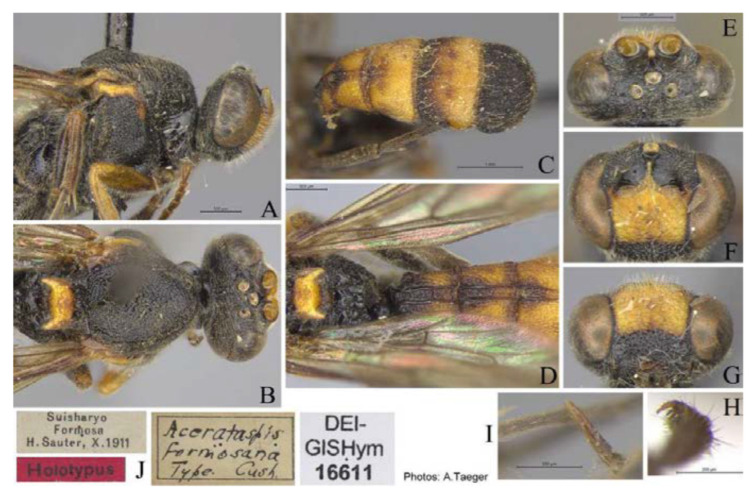
*Acerataspis formosana* Cushman, female. (**A**) Head and mesosoma, lateral view; (**B**) head and mesosoma, dorsal view; (**C**) T3–T6, dorsal view; (**D**) scutellum, metanotum, propodeum and T1–T3, dorsal view; (**E**) head, dorsal view; (**F**) head, anterior view; (**G**) head, anterior ventral view; (**H**) fore tarsal claw; (**I**) ovipositor, lateral view; (**J**) labels of holotype. (Images: A. Taeger).

**Figure 10 insects-14-00389-f010:**
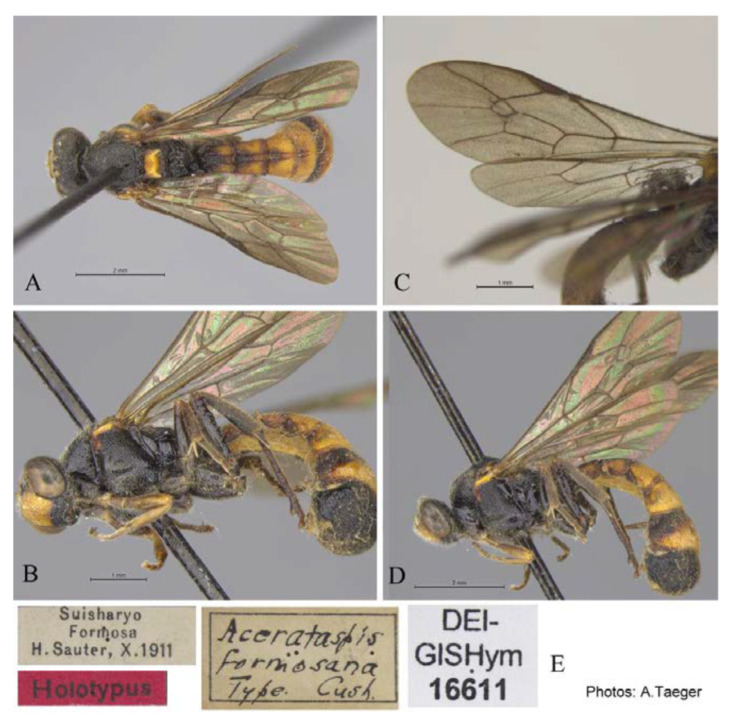
*Acerataspis formosana* Cushman, holotype, female. (**A**) Habitus, dorsal view; (**B**,**D**) habitus, lateral view; (**C**) wings; (**E**) labels. (Images: A. Taeger).

#### 3.4.4. *Acerataspis fukienensis* Chao, 1957 (Figure 11, Figure 12, Figure 13, Figure 14 and Figure 15)

*Acerataspis fukienensis* Chao, 1957, 7:105 [27]. Holotype female from China.

**Materials examined.** Holotype, 1♀, Label “HOLOTYPE; Shaowu, Fukien/China 1945.VIII-20/H.F. Chao Coll.; *Acerataspis fukienensis*/Chao, sp. nov./Holotype” (IOZ). **Other materials:** CHINA: 1**♀**1♂, Yunnan Province, Xishuangbanna, Menghai, Menghun Town, Area E1, rain forest, 21°44.944′ N 100°26.182′ E, 1677 m, 25.VI–15.VII.2021, Li Ma leg., Malaise trap, SCBG-E0000045, SCBG-E0000048 (SCAU).

**Diagnosis.** Area superomedia of propedeum medially with a transverse carina which divides the area into two parts: anterior transverse carina joining area superomedia at middle. Fore wing with 1cu-a opposite M&RS (Figure 12A), areolet petiolate, T4 with a pair of carinae on anterior half (Figure 12D).

**Description.** Female (holotype) (Figure 11). Body length 9.0 mm, antenna length 9.0 mm.

Head. Face and clypeus densely punctate, punctures on central area more or less transverse.

Mesosoma. Pronotum with dorsal 0.3 strongly punctate, lower part polished with several transverse short wrinkles. Epomia strong. Mesoscutum strongly punctate. Scuto-scutellar groove with 3short longitudinal carinae. Scutellum irregularly and coarsely punctate. Propedeum with lateromedian longitudinal carinae nearly parallel; anterior transverse carina joining area superomedia at middle; area superomedia separated from area basalis by the median abscissa of anterior transverse carina; area petiolaris punctate. Mesopleuron densely and coarsely punctate, and setose. Metapleuron polished, with minute punctures, setose, juxtacoxal carina weak.

Wings. Fore wing (Figure 12A) with 1cu-a opposite M&RS; areolet petiolate, receiving 2m-cu before middle. Hind wing with nervellus interrupted at lower 0.4, distal abscissa of CU weakly pigmented.

Legs. Fore and hind tarsal claw with three pectinae (Figure 12B,C).

Metasoma. Coarsely and densely punctate. T1–T3 with a pair of paralleled lateromedian longitudinal carinae. T1 and T2 rugose punctate, with central area between median carinae irregularly rugose. Lateromedian longitudinal carinae of T2 connected by a transverse carina beyond middle. T3 with a weak carina between the median carinae, punctate, transverse, T4 with a pair of lateromedian longitudinal carinae on anterior half (Figure 12D).

Colour (Figure 11). Antenna reddish brown, with apical flagellomeres blackish brown; scape largely black with apex dorsally reddish brown, ventrally yellow; pedicel basally black, apically reddish brown. Face largely yellow, with lower 0.3 black. Mandible brown. Tegula blackish brown. Subtegular ridge reddish brown. Scutellum black with apex yellowish brown. Fore coxa blackish brown, trochanter and femur dark brown (with ventral side lighter), tibia and tarsus yellowish brown. Mid and hind legs blackish brown, tarsus dark brown, spurs yellowish white. T1–T3 black, T4 with apical medially interrupted yellow band, T5 with apical margin dorsally medially yellow. Hind leg entirely blackish. Wings hyaline, pterostigma and veins blackish brown.

Variation (n = 1)**:** Scutellum black with yellowish-brown mark on apex to entirely black.

Male (Figure 13, Figure 14 and Figure 15). Body length 7.4 mm, fore wing length 5.5 mm.

Head. Face rugose punctate (Figure 13B), 1.3 times as high as wide. Frons rugulose dorsally and transversely wrinkled above antennal sockets. Malar space 0.5 times as mandibular basal width. POL:OD:OOL = 7:7:6. Antenna with 47 flagellomeres, distal 12 flagellomeres with sensillar plates on dorsal side, apical segment of antenna 2.0 times as long as its basal width.

Mesosoma. Pronotum (Figure 14A) in lateral view dorsally and posteriorly strongly punctate, centrally anteriorly with a small glabrous patch, and with several transverse wrinkles on lower part. Mesoscutum (Figure 14B) strongly and coarsely punctate. Scuto-scutellar groove with 3–4 short carinae. Scutellum strongly and coarsely punctate, with lateral carinae reaching to the apex of scutellum. Metanotum slightly convex with an M-shaped wrinkle. Mesopleuron (Figure 14A) strongly convex above sternaulus, strongly rugose punctate; epicnemial carina reaching to upper 0.6 of mesopleuron. mesopleural furrow glabrous. Metapleuron slightly convex, centrally punctulate, juxtacoxal carina fine and complete. Propodeum short (Figure 14C), weakly convex, area externa and area dentipara strongly rugose; area superomedia rugulose, weakly arched at middle, 0.9 times as long as wide, receiving anterior transverse carina near middle; area lateralis rugose; lateral longitudinal carina with anterior abscissa absent, spiracle round.

Wings. Fore wing (Figure 14D) with 1cu-a opposite M&RS, areolet sessile, and receiving 2m-cu before middle, hind wing with Cu & cu-a vein weakly interrupted at lower 1/3, distal abscissa of CU weakly pigmented with proximal section spurious.

Legs. Tarsal claws (Figure 15A,B) strongly pectinate, fore claw with 10 pectinae, mid and hind claws with 7–9 pectinae. Middle tibia with two spurs (Figure 15C). Hind femur 5.0 times as long as maximum width, the longer spur of hind tibia 0.5 times as long as basitarsus.

Metasoma (Figure 14F). Strongly and coarsely punctate. T1–T4 with a pair of median longitudinal carinae. Median longitudinal carinae of T1 and T2 reaching to posterior margin of T1 and T2, and closed posteriorly, median longitudinal carinae of T3 reaching to posterior 0.8 of tergite and posteriorly open; median longitudinal carinae of T4 fine and reaching to posterior 0.6 of T4, and with longitudinal wrinkles in between; area of median longitudinal carinae of T1–T3 irregularly sculptured. T5 and T6 strongly and closely punctate.

Colour. Body mainly black (Figure 13A), covered with white setae. Face largely yellow with lower 0.25 black. Mandible black. Palpi black. Antenna ventrally yellowish brown and dorsally blackish brown. Scutellum yellowish orange. Tegula blackish brown, mediolaterally tinged with yellowish brown. Subtegular ridge orange yellow. Metasoma black with orange marks on T4 and T5, T4 posteriorly with a irregular transverse yellowish orange band (posterior margin of T4 black), T5 with a narrow orange transverse band posteriorly. Fore leg blackish brown, trochanter with apical margin brownish yellow; femur anteriorly brownish yellow (proximal base of femur blackish brown), posteriorly black with proximal apex brownish yellow. Mid and hind legs black, with spurs light blackish brown. Wings hyaline, veins and pterostigma blackish brown.

**Distribution.** Oriental region.

**Comments.** Chao [26] stated that the holotype of *Acerataspis fukienensis* was deposited in ‘Fukien Agricultural College’ (=Fujian Agricultural and Forestry University), but the type materials of this species are currently deposited in the Institute of Zoology, Chinese Academy of Science (Beijing).

**Differential diagnosis**. This species is similar to *A. sinensis* and *A. szechuanensis* Chao,1962, but can separated from these species by: fore wing with 1cu-a opposite M&RS (fore wing of *A. sinensis* and *A. szechuanensis* with 1cu-a distinctly postfurcal to M&RS); areolet petiolate (areolet of *A. sinensis* and *A. szechuanensis* sessile); fore and hind claws with three stout pectinae (with six fine pectinae in *A. sinensis* and *A. szechuanensis*); T4 with a pair of carinae on anterior half (T4 of *A. sinensis* without carina, of *A. szechuanensis* with a single short carina anteriorly); T4 with apical yellow narrow band medially interrupted (T4 of *A. sinensis* and *A. szechuanensis* with apical yellow band complete and largely broadly yellow); anterior transverse carina of propodeum joining area superomedia near middle (both *A. sinensis* and *A. szechuanensis* with anterior transverse carina joining area superomedia near posterior transverse carina).

**Figure 11 insects-14-00389-f011:**
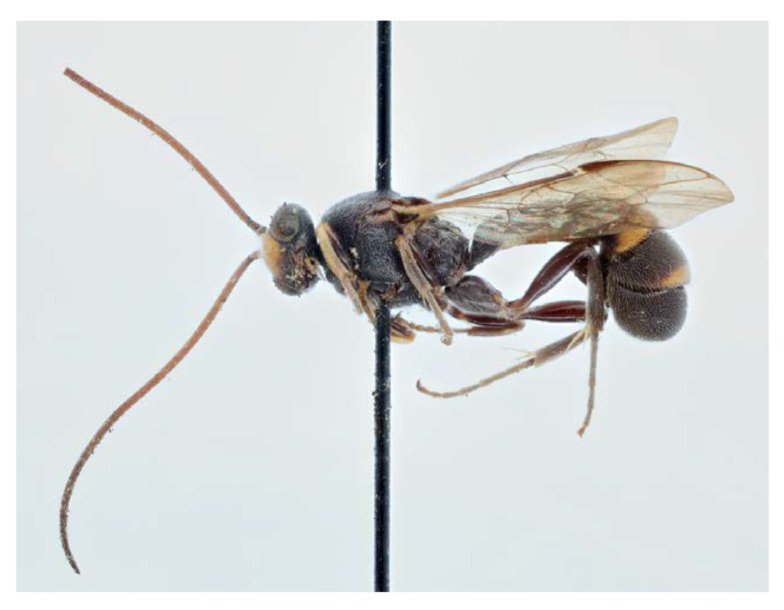
*Acerataspis fukienensis* Chao, holotype female, habitus, lateral. (Image: Huang Zhengzhong).

**Figure 12 insects-14-00389-f012:**
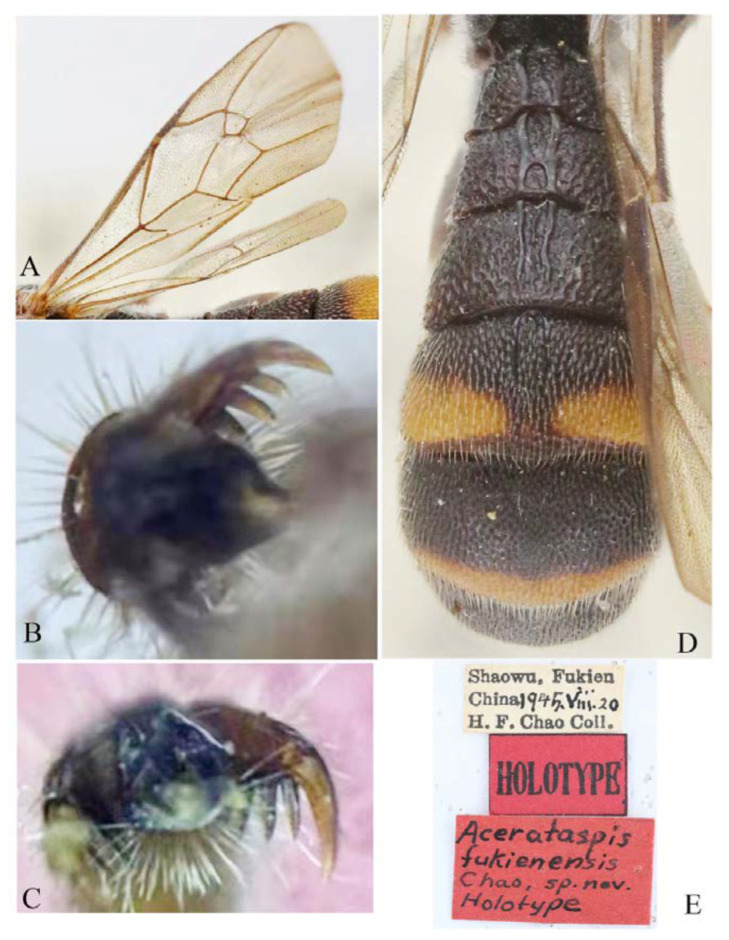
*Acerataspis fukienensis* Chao, holotype female. (**A**) Fore wing; (**B**) fore tarsal claw; (**C**) hind tarsal claw; (**D**) metasoma, dorsal view; (**E**) labels (Image: Huang Zhengzhong).

**Figure 13 insects-14-00389-f013:**
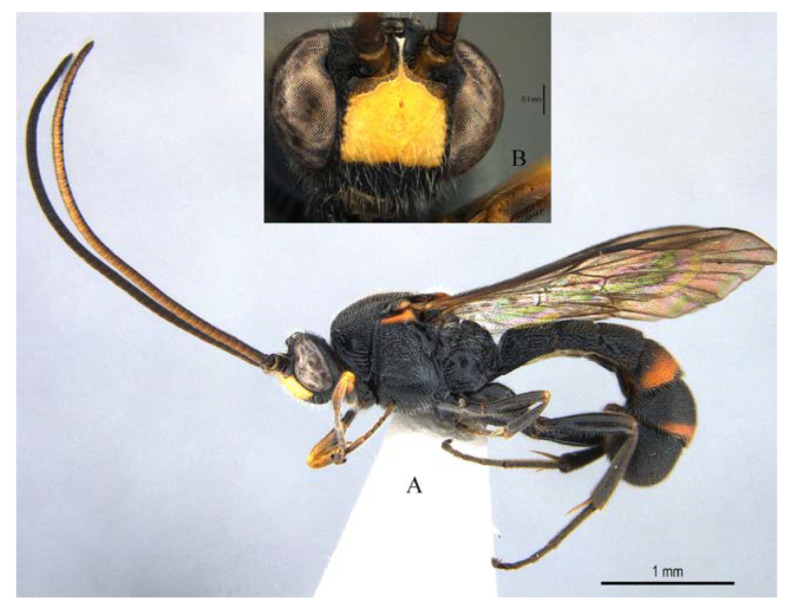
*Acerataspis fukienensis* Chao (non-type specimen ID: SCBG-E0000045), male. (**A**) habitus, lateral view; (**B**) head, frontal view.

**Figure 14 insects-14-00389-f014:**
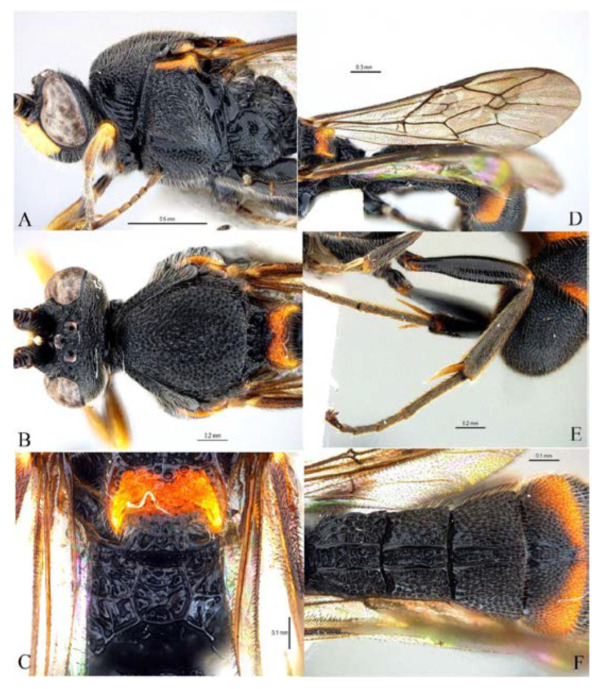
*Acerataspis fukienensis* Chao (non-type specimen ID: SCBG-E0000045), male. (**A**) Head and mesosoma, lateral view; (**B**) head and mesosoma, dorsal view; (**C**) scutellum, metanotum and propodeum, dorsal view; (**D**) fore wing; (**E**) hind leg; (**F**) T1–T4, dorsal view.

**Figure 15 insects-14-00389-f015:**
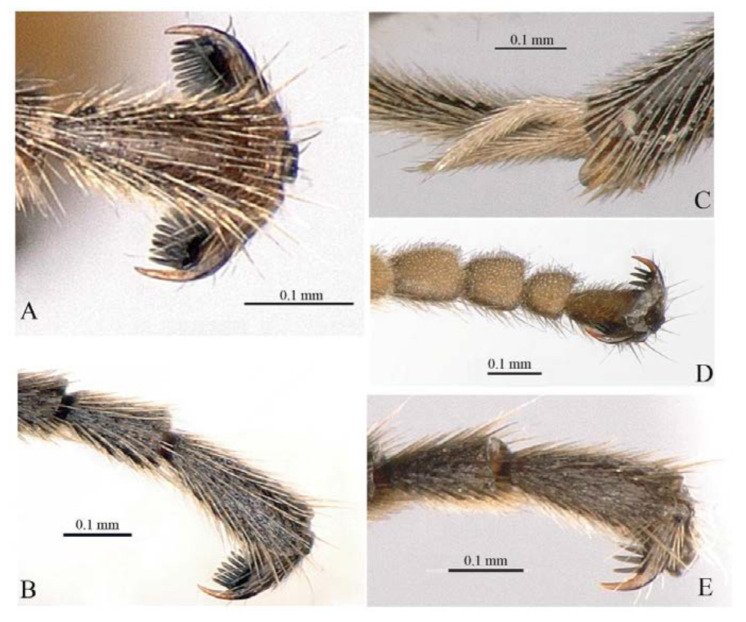
*Acerataspis fukienensis* Chao (non-type specimen ID: (**A**–**C**), SCBG-E0000045, (**D**–**F**), SCBG-E0000048). (**A**) Fore claw of male; (**B**) hind tarsus of male; (**C**) middle tibial spurs of male; (**D**) fore tarsal claw of female; (**E**) hind tarsus of female.

#### 3.4.5. *Acerataspis fusiformis* (Morley, 1913) (Figure 16, Figure 17, Figure 18, Figure 19 and Figure 20)

*Metopius fusiformis* Morley, 1913: 268 [1]. Holotye female from Myanmar.

**Materials examined.** Holotype: 1♀, labelled “Type C.M; Carin Cheba 900–1100 m, L.Fea V XII-88; named by Claude Morley, *Metopius fusiformis*, ♀, ii.1913” (MCSN). Approximate coordinates 19°19′ N 96°48′ E (Fea 1888). Examined by photos provided by Maria Tavano (Museum of Natural History “Giacomo Doria” (Museo Civico di Storia Naturale Giacomo Doria) Genova, Italy). **Non-type materials:** CHINA: 1♀, Hong Kong, Ping Shan Chai, 22°29′8.30″ N 114°11′14.74″ E, alt. 140 m, Malaise trap (M335), 17–31.III.2018, Christophe Barthélémy leg., SCAU-3013715 (HKU); 1♂, Hong Kong, Ping Shan Chai, 22°29′8.30″ N 114°11′14.74″ E, alt. 140 m, Malaise trap (M338), 31.III–14.IV.2018, Christophe Barthélémy leg., SCAU-3013716 (HKU); 1♂, Hainan Province, Nankai Town, LSX-727, 19°4’20.77″ N,109°22’28.2″ E, HN9, 30.IV–30.V.2020, Malaise trap, SCAU-3011024 (SCAU); 1♀, Hainan Province, Nankai Town, 19°4′43.32″ N,109°23′38.74″ E, HN10, 30.IV–30.V.2020, Malaise trap, Luo Shi-Xiao leg., SCAU-3011028 (SCAU); 1♂, Yunnan, Xishuangbanna Botanical Garden, rubber forest, 21°52′ N, 101°19’ E, alt. 543 m, 16.V–19.VI.2021, Malaise trap, Li Ma leg., SCBG-E0000043 (SCAU). THAILAND: 1♂, Chiang Mai, Mae Taeng, Pa Pae, 19°14′30.6″ N, 98°39′14.1″ E, old sec. forest with *Camellia sinensis* var. *assamica*, Malaise trap (Dara#1), 04.V–25.V.2017 (#5), Monsoon Tea leg. SCAU-3013717 (QSBG).

**Diagnosis.** Fore wing with 1cu-a opposite M&RS, areolet sessile (Figure 17 and Figure 18). Propodeum with costa joining area superomedia at middle (Figure 19C). Fore tarsal claw of female with three pectinae (Figure 19F). T1–T3 with a pair of median longitudinal carinae, with a weak longitudinal wrinkle in between. T4 without median carina. Antenna with scape entirely yellow. Metanotum yellow. Mid femur black with proximal base and apex yellow. Hind tibia with basal half yellow.

**Description.** Female. Body length 8.0–11.0 mm, fore wing length 7.0–7.8 mm.

Head. Face rugose punctate (Figure 17B), 1.3 times as high as wide, antenna with 60 flagellomeres. Frons with minute punctures on dorsal half, smooth on lower half. Mandible with upper tooth longer than lower tooth, outer margin of upper tooth weakly convex. Malar space 0.3 times width of mandibular base. POL:OD:OOL = 10:10:6.

Mesosoma (Figure 19A,B). Pronotum polished and transversely crenulate on lower half, dorsally densely punctate and setose. Epomia weak. Mesoscutum densely and coarsely punctate. Scutellum transverse, shallowly punctate, moderately densely setose. Propodeum (Figure 19C) with anterior transverse carina joining area superomedia on anterior 0.6, area superomedia slightly convergent posteriorly from the meeting point with anterior transverse carina; area externa punctate and setose; area dentipara with sparse punctures, laterally punctate along lateral longitudinal carina; area lateralis rugulose and densely setose; area postero-externa rugulose; spiracle oval and touching pleural carina. Mesopleuron punctate reticulate (Figure 19A), with speculum and posterior margin along mesepisternum glabrous and impunctate, mesopleuron strongly convex above sternaulus; epicnemial carina sharp and well defined on lower 0.4 (below the sternaulus) and obscure on upper 0.6 (the convex part of mesopleuron above sternaulus); sternaulus broadly impressed. Metapleuron sparsely punctulate, setose.

Wings. Fore wing with 1cu-a opposite to M&RS (Figure 19D), areolet sessile and receiving 2m-cu before middle. Hind wing with nervellus interrupted at lower 0.3, distal abscissa of CU weakly pigmented.

Legs. All tarsal claws with three pectinae (Figure 19F–H), fore tarsal claw with the middle pecten longer than the other two pectinae. Longer spur of hind tibia 0.67 times length of basitarsus. Hind femur 4.1 times its maximum width, 1.0 times the length of tibia.

Metasoma. Strongly punctate reticulate. T1–T3 with a pair of lateromedian longitudinal carinae, area between lateromedian longitudinal carinae rugose, or sometimes with irregular longitudinal wrinkles. T1 without dorsolateral carina. T3 with lateromedian longitudinal carinae reaching to posterior 0.88. T4 without lateromedian longitudinal carinae. Anterior lateral corner of T4–T6 around spiracle with a distinct longitudinal impression. Ovipositor 0.72 times as long as hind tibia. Ovipositor sheath slender, sparsely setose.

Colour (Figure 17A). Body black with yellow marks. Face with upper 0.6 yellow and lower 0.3 black. Mandible dark brown with extreme base black. Palpi whitish yellow. Scape yellow with laterally outer side black, pedicel dorsally black and ventrally yellow, flagellum dorsally blackish brown, ventrally fuscous on basal 0.6, and gradually becoming blackish brown on apical 0.3. Pronotum black with a yellow spot on posterior lateral corner. Tegula, parategula, scutellum, subtegular ridge and metanotum entirely yellow. Fore coxa dark brown with apex yellow, fore trochanter yellow, fore femur blackish on basal half and yellow on apical half, fore tibia and tarsus yellow; mid coxa blackish on basal 0.6 and yellow on apical 0.3, mid trochanter yellow, mid femur black with apical 0.2 yellow, mid tibia and tarsus yellow; hind coxa black, hind trochanter basally fuscous and apically yellow, hind femur black with extreme basal 0.1 whitish yellow, hind tibia with proximal 0.6 yellow (except the extreme base black) and apical 0.4 black, hind tarsus with 1–4 segments dark fuscous, 5th segment brownish yellow (Figure 19E). Wings hyaline, wing base yellow, veins and pterstigma blackish brown. T1 laterally largely yellow, basal 0.3 and area between lateromedian longitudinal carinae black, T2 and T3 black with a pair of moderately large yellow marks on posterior lateral corner (extreme apical margin of T2 and T3 black), T4 (except for extreme apical margin black) and T5 apically with a yellow band, T6 black. Ovipositor sheaths fuscous.

Male (Figure 18). Body length 8.6 mm, fore wing length 6.8 mm. Similar to female, except: propodeum shorter than female (Figure 20D); fore and middle tarsal claws densely pectinate, with 9–12 pectinae (Figure 20F), hind claw with 3–4 pectinae (Figure 20G). Scape dorsally black and ventrally yellow. T1–T3 posteriorly with a pair of separated yellow marks (Figure 20H,I). Fore wing with 1cu-a opposite M&RS.

**Distribution**. Oriental region (newly recorded from Hong Kong, Yunnan and Thailand).

**Comments**. The head and antenna of the type was separated into two parts. We reviewed historical literature of the field notes of Leonardo Fea, who collected the holotype specimen of *A. fusiformis* during his travels to the Karen Hills of Bristish Burma [27,28]. The Karen Hills, or Kayah-Karen Mountains, is a large highland area in eastern Myanmar stretching from Shan Mountain southward to Tenassurin Mountain, although Fea had defined the Karen Hills as a restricted area between two tributaries of the Sittaung River (Thauk Ye Kupt and Paunglaung rivers) and the Nam Pawn River, a tributary of the Salween River. We reconstructed his sampling sites using his field diaries and supplementary maps [27,28] and estimate the approximate type locality of *A. fusiformis* to be 19°19′ N 96°48′ E.

There are some variations between some female specimens (n = 1, specimen ID: SCAU-3013715), propodeum with area dentipara densely punctate, lateromedian longitudinal carinae subparallel, area petiolaris transversely wrinkled with punctures in between. Yellow marks on T1–T4 larger (Figure 17), T1 largely yellow except for black anterior base, T2 with yellow marks not interrupted at middle (Figure 17). The pecten number of hind tarsal claws is different between left leg and right leg in one specimen (n = 1, specimen ID: SCAU-3011028) from Hainan Province of China (three pectinae on left hind claw and four pectinae on the right one). Variations in male specimens as follows: in SCAU-3013716, T4 with apical yellow band interrupted at middle (Figure 18 and Figure 20H); in No. SCAU-3013717, with 1cu-a postfurcal to M&RS) (Figure 20E).

Due to the brief original description of *A. fusiformis* and similar colour patterns of *A. fusiformis* and *A. clavata*, many specimens of *A. fusiformis* from different regions in previous studies [5,10,13,14,29,30,31] were incorrectly determined as *A. clavata* (Uchida, 1934). However, their *CO1* genetic distances between *A. fusiformis* and *A. clavata* were 11–12% (Appendix A). Therefore, their previous identification should be checked by using the main differential diagnostic characters: fore claw with three visible pectinae, fore wing with 1cu-a opposite to M&RS, and scutellum yellow in *A. fusiformis*; while in *A. clavata*, tarsal claw strongly pectinated, fore wing with 1cu-a postfurcal to M&RS, and scutellum black.

It is necessary to mention both *A. fusiformis* and *A. clavata* were collected in the same sampling sites in Thailand and Hong Kong. In Northern Thailand, the sampling site was in an old secondary *Decteracarpus* forest with *Camellia sinensis* var. *assamica* in the understory. There are similar communities with *Camellia sinensis* var. *assamica* around the type locality of *A. fusiformis* on the Karen Hills at Leonardo Fea’s sampling sites [32].

**Figure 16 insects-14-00389-f016:**
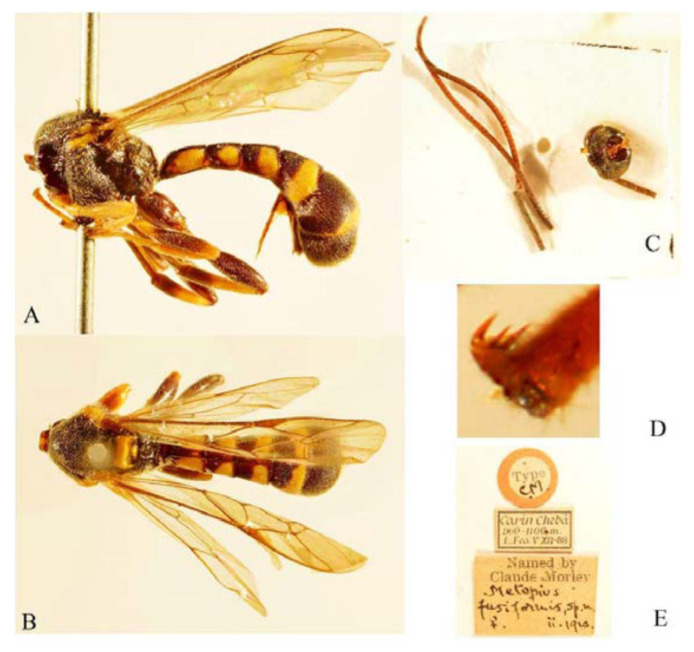
*Acerataspis fusiformis* (Morley), holotype, female. (**A**) Habitus, lateral view; (**B**) habitus, lateral view; (**C**) head and antenna; (**D**) fore tarsal claw; (**E**) labels. (Images: Maria Tavano).

**Figure 17 insects-14-00389-f017:**
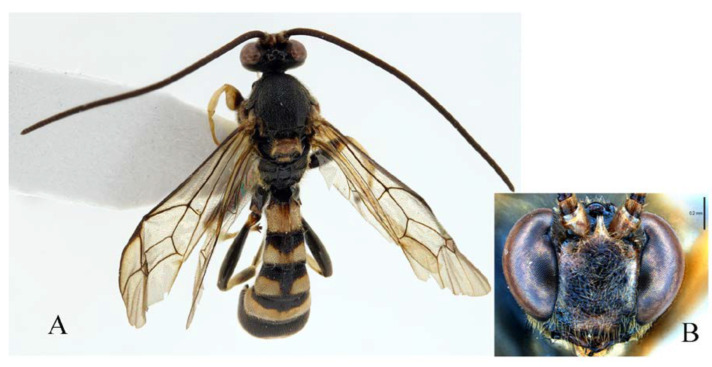
*Acerataspis fusiformis* (Morley) (non-type, specimen ID: SCAU-3013715), female. (**A**) habitus, dorsal view; (**B**) head, frontal view. (Note: Facial colouration was altered after the DNA extraction experiment; the original colour of the species is yellow).

**Figure 18 insects-14-00389-f018:**
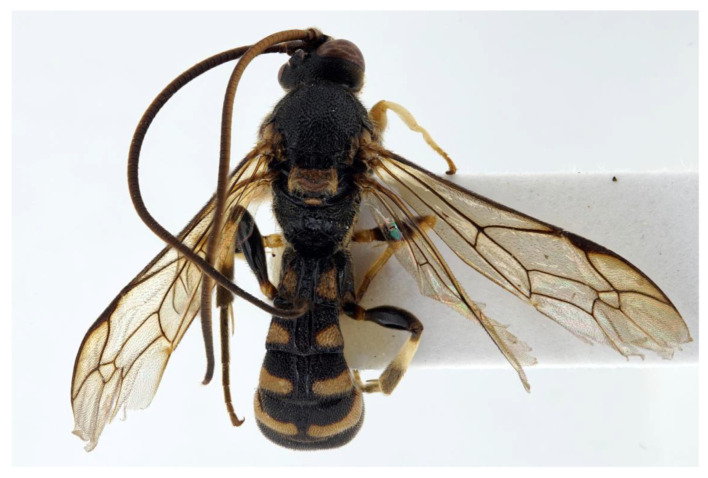
*Acerataspis fusiformis* (Morley) (non-type, specimen ID: SCAU-3013716), male, dorsal view.

**Figure 19 insects-14-00389-f019:**
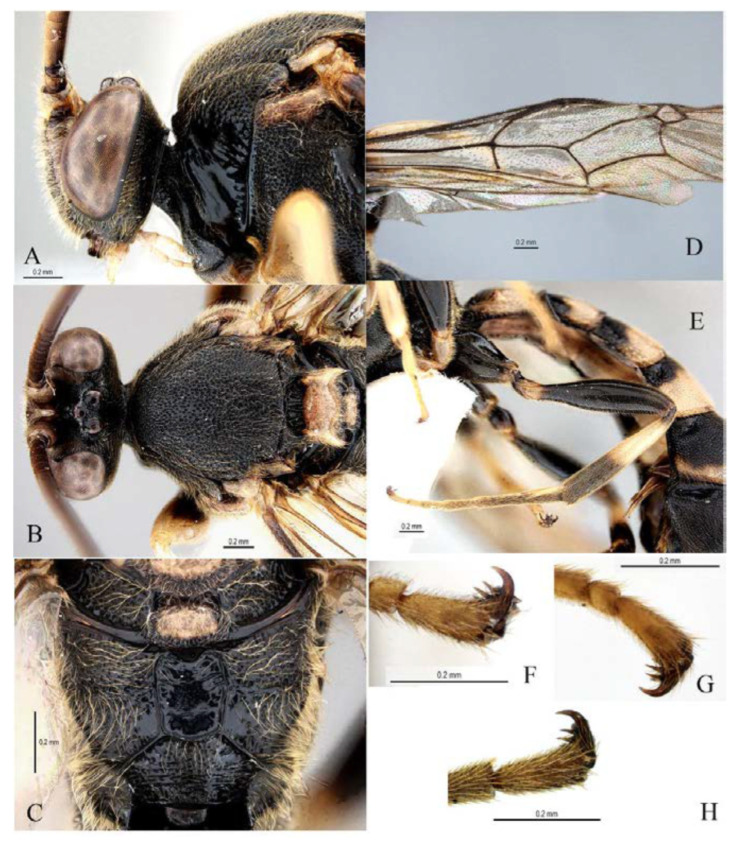
*Acerataspis fusiformis* (Morley) (non-type, specimen ID: SCAU-3013715), female. (**A**) Head and anterior part of mesosoma, lateral view; (**B**) head and mesosoma, dorsal view; (**C**) metanotum and propodeum, dorsal view; (**D**) fore wing veins; (**E**) hind leg; (**F**) fore tarsal claw; (**G**) mid tarsal claw; (**H**) hind tarsal claw.

**Figure 20 insects-14-00389-f020:**
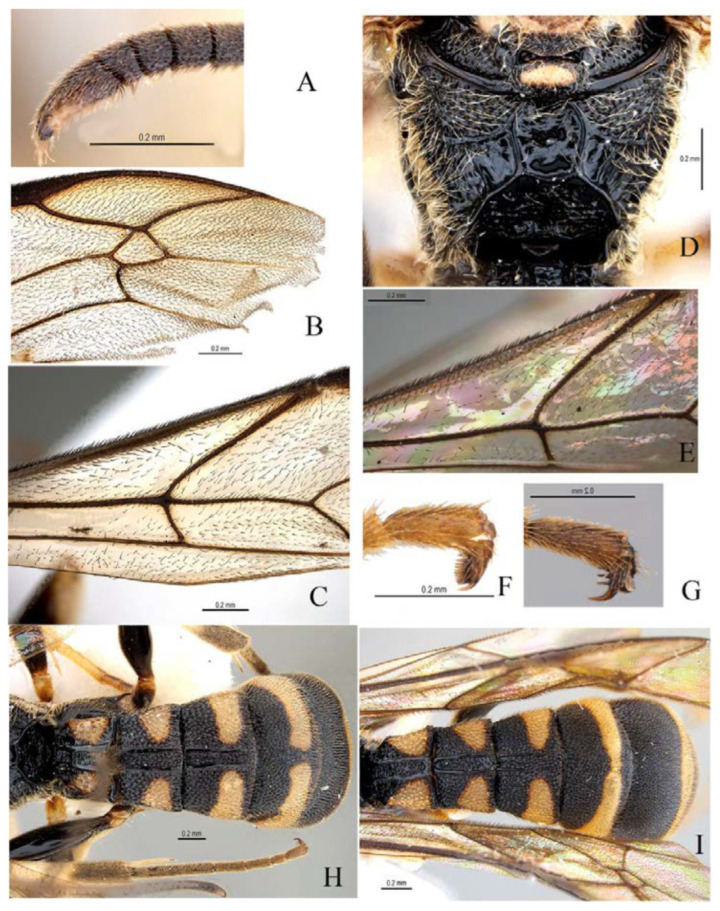
*Acerataspis fusiformis* (Morley) (non-type, specimen ID: SCAU-3013716, SCAU-3013717), male. (**A**) Apical part of flagellum; (**B**,**C**,**E**) fore wing veins ((**B**,**C**), SCAU-3013716, E, SCAU-3013717); (**D**) metanotum and propodeum, dorsal view; (**F**) fore tarsal claw; (**G**) hind tarsal claw; (**H**) metasoma with apical transverse band on T4 separated, dorsal view (SCAU-3013716); (**I**) metasoma with apical transverse band complete (SCAU-3013717).

#### 3.4.6. *Acerataspis maliae* Liu & Chen sp. nov. (Figure 21, Figure 22 and Figure 23)

**Materials examined.** Holotype, 1♀, CHINA: Yunnan, Xishuangbanna, Menghai, Menghun Town, alt. 1,621 m, Area E1, rainforest, Malaise trap, 21°44.944′ N 100°26.182′ E, 25.VI–15.VII.2021, Li Ma leg., SCBG-E0000047 (SCAU). Paratype. CHINA: 1♀, Yunnan Province, Xishuangbanna, Menghai, Bulangshan Village, alt. 1706 m, Area A2, forest, 21°45.061′ N 100°21.661′ E, 15.IX–15.X.2021, Malaise trap, Li Ma leg., SCBG-E0000042 (SCAU).

**Diagnosis.** Fore and mid tarsal claws with four pectinae, hind claw with seven pectinae; fore wing with 1cu-a slightly postfurcal to M&RS, areolet weakly sessile above. Scape yellow with a black stripe laterally (Figure 22A).

**Description.** Holotype. Female. Body length 7.7 mm, fore wing length 5.7 mm.

Head. Face strongly rugose punctate (Figure 21B), 1.3 times as high as wide. Malar space 0.44 times as long as the basal width of mandible. Frons dorsally densely punctulate, and glabrous above antennal sockets. POL:OD:OOL = 11:9:8. Antenna with 58 flagellomeres.

Mesosoma. Pronotum (Figure 22B) dorsal 0.3 strongly punctate, anteriorly centrally with a glabrous patch, and with several transverse short carinae on lower 0.3. Mesoscutum strongly punctate (Figure 22C), punctures on central area somewhat sparser than those on lateral and posterior areas. Scuto-scutellar groove with three short longitudinal carinae. Scutellum sparsely and coarsely punctate (Figure 22D). Epicnemium anteriorly crenulated. Epicnemial carina reaching to upper 0.5 of mesopleuron. Mesopleuron (Figure 22B) strongly convex above sternaulus, strongly punctate reticulate, mesopleural furrow weakly crenulated. Metapleuron (Figure 23F) sparely punctulate, area between punctures polished, juxtacoxal carina indicated by medially interrupted wrinkles. Propodeum (Figure 22D) with areas rugose punctate, area superomedia weakly wrinkled, polished, as long as its maximum width, and receiving anterior transverse carina at middle; lateral longitudinal carina with anterior abscissa weak; area lateralis rugulose. area postero-externa strongly rugose.

Wings. Fore wing (Figure 23D) with 1cu-a slightly postfurcal to M&RS, areolet slightly sessile above, receiving 2m-cu before middle. Hind wing with nervellus interrupted at middle (Figure 23E), distal abscissa of CU distinct.

Legs. Fore tarsal claw with four pectinae (Figure 23A), mid tarsal claw with four pectinae (Figure 23B), hind tarsal claw with six pectinae (Figure 23C), pectinae of hind tarsal claw of different size (Figure 23C).

Metasoma (Figure 22E,F). Strongly rugose punctate. T1–T3 with a pair of lateromedian longitudinal carinae, T1 transversely wrinkled between cariane, T2 rugulose between carinae and with a wealy defined longitudinal wrinkle between the carinae, T3 centrally with a weak longitudinal carina between the lateromedian longitudinal carinae (Figure 22 E). T4 rugose punctate, medially with irregular longitudinal wrinkles (Figure 22F). Ovipositor sheath 0.58 times as long as hind basitarsomere.

Colour (Figure 21A). Face mostly yellow (centrally brownish yellow) with lower 0.25 black. Mandible basally and apically black, medially brown. Palpi yellowish brown. Antenna dorsally fuscous, ventrally yellowish brown, scape (Figure 22A) yellow with a lateral black stripe, pedicel and base of first flagellomere black. Tegula and subtegular ridge yellowish brown. Scutellum yellow with anterior 0.3 yellowish brown. Metasoma mainly black, T1–T3 posterior laterally with brownish yellow triangular spots, centrally interrupted by the lateromedian longitudinal carinae (Figure 22E,F), T4–T5 posteriorly with a transverse brownish yellow band. Fore leg with coxa black, trochanter with outer side black and inner side brownish yellow, femur fuscous, outer side with basal 1/2 black, tibia and tarsus brownish yellow. Middle leg with coxa, trochanter and femur black, tibia fuscous with basal 0.4 brownish yellow, tarsus dark brown. Hind leg (Figure 21) with coxa, trochanter and femur black, tibia blackish brown with proximal 0.3 brown, tarsus brown. All tibial spurs whitish yellow. Wings hyaline, veins blackish brown. Ovipositor sheath brownish yellow.

**Male.** Unknown.

**Distribution.** Oriental region (China).

**Etymology.** The species is named after Prof. Li Ma, a hymenoptera taxonomist at Yunnan Agricultural University.

**Differential diagnosis.** The colour of the new species is very similar to *A. clavata* and *A. fusiformis*. It can be separated from *A. clavata* by fore and mid tarsal claws with four pectinae (fore and middle claws with more than six pectinae in *A. clavata*) and scape yellow dorsally (scape dorsally black in *A. clavata*). It differs from *A. fusiformis* by hind tarsal claw with seven pectinae (hind claw with three or four pectinae in *A. fusiformis*). The status of the new species is also supported by *COI* gene sequence; the genetic distances between this species and *A. clavata* and *A. fusiformis* range from 12.9% to 14.4%. (Appendix A)

**Figure 21 insects-14-00389-f021:**
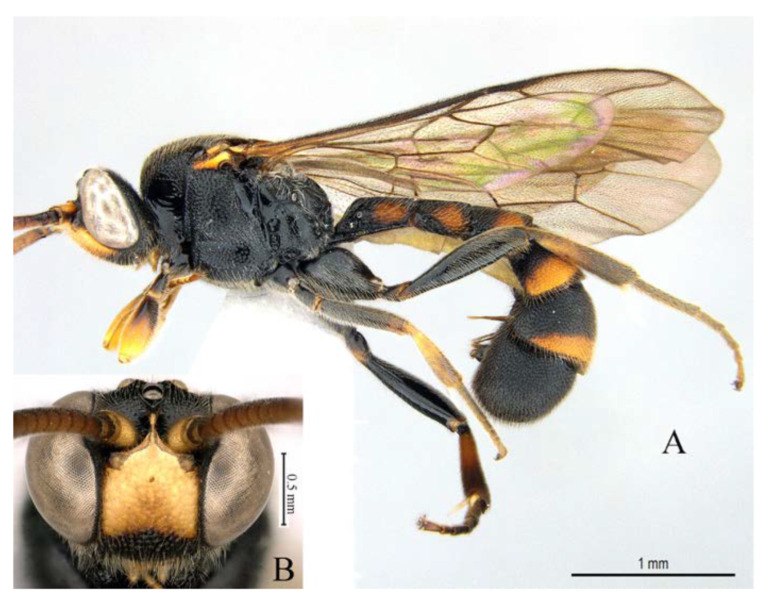
*Acerataspis maliae* Liu & Chen, sp. nov. holotype. (**A**) habitus, lateral view; (**B**) head frontal view. (Note: Facial colouration was altered after the DNA extraction experiment; the original colour of all species is yellow).

**Figure 22 insects-14-00389-f022:**
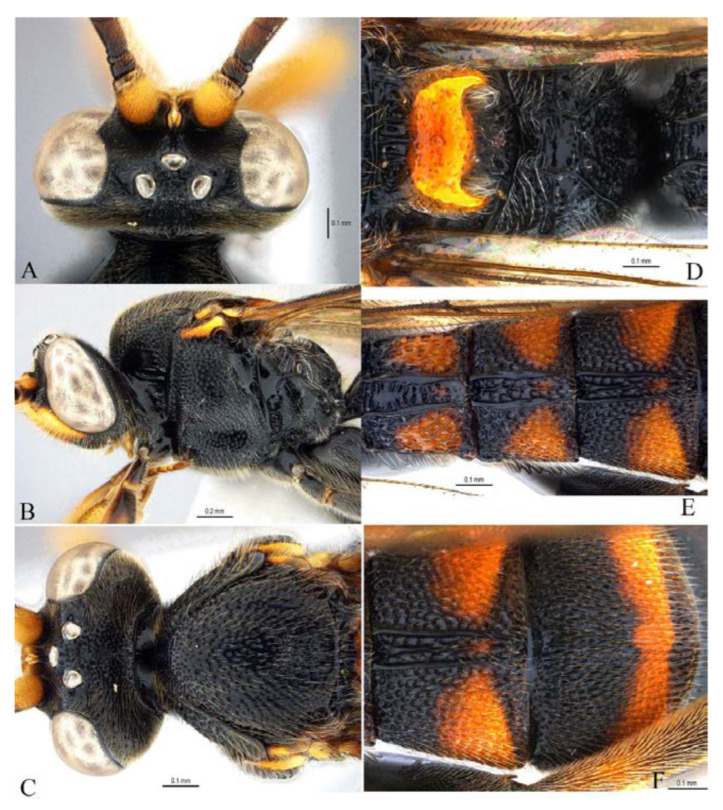
*Acerataspis maliae* Liu & Chen, sp. nov. holotype, female. (**A**) Head, dorsal view; (**B**) head and mesosoma, lateral view; (**C**) head and anterior part of mesosoma, dorsal view; (**D**) scutellum, metanotum and propodeum, dorsal view; (**E**) T1–T4, dorsal view; (**F**) T3–T4 dorsal view.

**Figure 23 insects-14-00389-f023:**
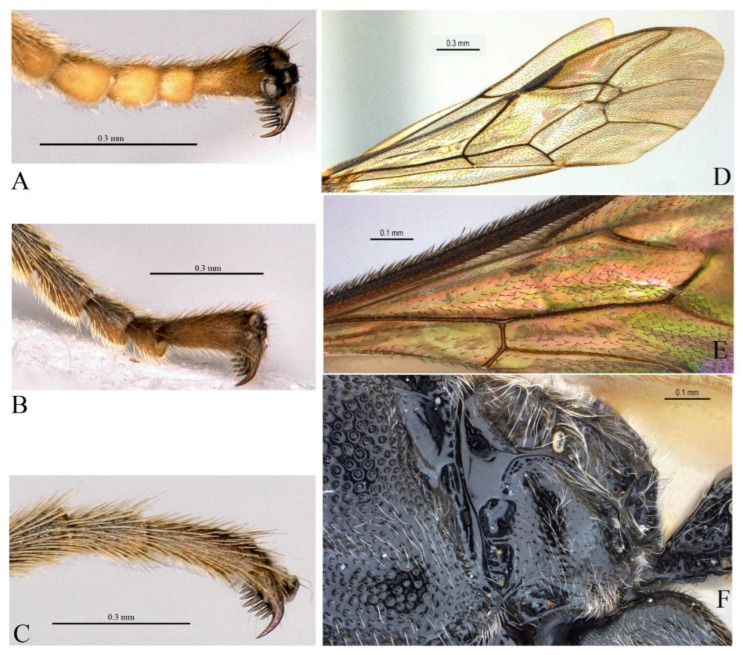
*Acerataspis maliae* Liu & Chen, sp. nov. holotype, female. (**A**) Fore tarsus; (**B**) middle tarsus; (**C**) hind tarsus; (**D**) fore wing; (**E**) hind wing; (**F**) posterior part of mesopleuron and metapleuron, lateral view.

#### 3.4.7. *Acerataspis separata* Liu, Reshchokv & Chen, sp. nov. (Figure 24, Figure 25, Figure 26, Figure 27, Figure 28 and Figure 29)

**Materials examined.** Holotype, 1♀, CHINA: Yunnan Province, Xishuangbanna, Menghai County, Bu Lang Shan Town, grass land, site A. 21°44.981′ N, 100°26.907′ E, alt. 1710 m, 15.VIII.2018, Li Ma leg., Malaise trap, SCAU-3011982 (SCAU). Paratype. 1♂, China: Yunnan Province, Xishuangbanna, Menghai County, Bu Lang Shan Town, grassland, site D, 21°44.745′ N, 100°26.070′ E, alt. 1621 m, Li Ma leg., Malaise trap, 16.VII–14.IX.2018, SCAU-3011978 (SCAU); 4♂, Yunnan Province, Xishuangbanna Botanical Garden, rubber forest, alt. 543 m, 21°912.439′ N 101°26.84′ E, 16.III–11.IV.2021, Li Ma leg., Malaise trap, SCBG-E0000044, SCBG-E0000051, SCBG-E0000052, SCBG-E0000054 (SCAU); 1♂, Yunnan, Xishuangbanna, Menghai, Bulangshan Village, alt. 1,621 m, Area E1, forest, malaise trap, 21°44.944′ N 100°26.182′ E, 15.IX–15.X.2021, Li Ma leg., Malaise trap, SCBG-E0000050 (SCAU).

**Diagnosis.** Fore wing (Figure 26D) with 1cu-a postfurcal to M&RS, areolet of fore wing sessile above. Propodeum with anterior transverse carina joining area superomedia at middle (Figure 25D). All tarsal claws densely pectinate. T1–T4 with yellow marks posteriorly, T4 with apical transverse band interrupted dorso-medially (Figure 25F).

**Description.** Female holotype. Body length 5.7 mm, fore wing length 4.3 mm.

Head. Face (Figure 24B) rugose punctate, punctures closed and forming some transverse wrinkles, combined face and clypeus 1.3 times as high as wide. Mandible slightly twisted, with upper tooth longer than lower tooth, mandible with a flange-like projection subasally. Malar space 0.5 times the width of mandibular base. Head 1.6 times as wide as long. Gena strongly narrowed behind eyes. POL:OD:OOL = 6:6:5. Antenna with 47 flagellomeres, first flagellomere 1.8 times as long as its apical width, 1.4 times as long as the second. Occiput strongly punctate and setiferous. Occipital carina complete.

Mesosoma. Pronotum (Figure 25A) with dorsal 0.3 strongly punctate and setiferous; lower 0.6 polished and glabrous, crenulate, with several transverse wrinkles along the posterior margin of pronotum. Epomia weak. Mesoscutum (Figure 25B) densely and closely punctate, and setose; posterior lateral corner of scutum with a lamella flange just behind the tegula that is 1.2 times the length of the tegula. Scuto-scutellar groove deeply crenulate, with four short longitudinal carinae. Scutellum transverse, punctate, with lateral posterior corner projected. Propedeum (Figure 25D) with area superomedia weakly anteriorly curved outward, polished and glabrous; anterior transverse carina joining area superomedia at middle; area externa sparsely punctate and with inner posterior corner polished; area dentipara with outer 0.6 rugose punctate and setose, inner 0.3 polished and glabrous; area lateralis weakly rugose, setose. Spiracle oval, connected to pleural carina. Mesopleuron (Figure 25C) strongly convex, densely and strongly punctate; sternaulus broadly impressed. Epicnemial carina reaching to upper 0.7 of mesopleuron; mesopleuron with a narrow triangular polished area along the position of mesopleural furrow (which is not indicated) (Figure 25C). Metapleuron weakly convex, with minute setiferous punctures, polished in between, anterior margin strongly crenulate, juxtacoxal carina complete, upper side with 1–2 wrinkles. Submetapleural carina complete, anteriorly with a projection.

Wings. Fore wing (Figure 26D) with 1cu-a postfurcal to M&RS, areolet sessile, ratio of 2rs-m and 3rs-m = 10:8, areolet receiving 2m-cu before middle. Hind wing (Figure 26E) with 12 hamuli, nervellus interrupted at lower 0.3, distal abscissa of CU weakly pigmented.

Legs. Fore femur swollen, 2.5 times as long as its maximum width; fore tarsal claw (Figure 26A) densely pectinate, with seven pectinae, basitarsus 0.8 times the combined length of tarsomeres 2–5. Middle femur 3.2 times as long as its maximum width, middle claw densely pectinate (Figure 26B). Hind femur (Figure 25E) 4.0 times as long as its maximum width, the longer tibial spur 0.7 times the length of basitarsus. Hind tarsal claw densely pectinate, teeth of equal size (Figure 26C).

Metasoma (Figure 25F). Strongly punctate. T1–T3 with a pair of latero-median carinae; area between latero-median carinae of T1 weakly transversely rugose, that of T2 weakly rugose, and with a longitudinal wrinkle reaching to posterior 0.8 of T2, that of T3 weakly rugose and with a longitudinal wrinkle reaching to posterior 0.8 of T3. Dorso-lateral carina of T1 absent. T4 densely punctate, medially longitudinally convex, without carina. T1 1.3 times as long as its apical width, T2 0.7 times as long as its apical width, T3 0.5 times as long as its apical width. T5 and T6 strongly punctate reticulate. Ovipositor sheath as long as hind basitarsus.

Colour. Body black with white maculations (Figure 24A). Face with upper 0.8 white (color altered after DNA extraction), lower 0.2 black. Mandible black. Palpi whitish. Antenna black, inner side of scape whitish. Tegula and subtegular ridge white. Scutellum white. Metanotum black. Fore coxa black with anterior side white; trochanter white, dorsally with a brown mark on basal 0.6; fore femur anteriorly white, posteriorly dark brown with apex white; fore tibia white, inner side with basal 0.7 dark brown; tarsus dark brown except for white basitarsus. Middle coxa black with white spot on anterior side; mid trochanter white, dorsally with basal 0.8 dark brown; mid femur black except for proximal base and extreme apex white; mid tibia white, inner side with basal 0.6 dark brown, mid tarsus with basitarsus mostly white (extreme apex dark brown), 2nd–5th tarsomeres dark brown. Hind coxa black, trochanter black with anterior side dark brown, hind femur black with proximal base white; tibia with proximal base and apical half black, basal half white; tarsus black. All tibial spurs white. T1 with a pair of large triangular white marks laterally; T2 with a pair of white marks on lateral posterior half; T3 with a medially interrupted subapical transverse white band; T4 with a medially interrupted subapical transverse white band; T5 with a complete transverse white band subapically. Wings hyaline, veins and pterostigma blackish brown.

Male (Figure 27, Figure 28 and Figure 29). Similar to female, except: body length 7.1 mm, fore wing length 5.4 mm. Epicnemial carina strong, reaching to upper end of mesopleuron (Figure 28B). Propodeum (Figure 28E) with area dentipara strongly rugose, area superomedia rugulose. Tarsal claws strongly pectinate (Figure 29A–C). White marks on T1–T3 smaller than female (Figure 28F). Scutellum white.

**Distribution.** Oriental region (Yunnan province of China).

**Etymology.** The specific name is derived from Latin *separatus*, referring to the separated maculation of T4.

**Comments.** This species can be separated from other congeneric species mainly by combinations of: T4 with separated maculations, densely pectinate claws, and fore wing with 1cu-a postfurcal to M&RS. This new species and *A.clavata* from Thailand (SCAU-3013718) share the separated maculation of T4, with differences between them as follows: *A. separata* sp. nov. with scape dorsally black (scape dorsally yellow in Thailand *A. clavata*) and metapleuron with three foveae on anterior margin (metapleuron with only two foveae in Thailand *A. clavata*). This new species was also supported by the interspecific distance of the *CO1* gene.

**Figure 24 insects-14-00389-f024:**
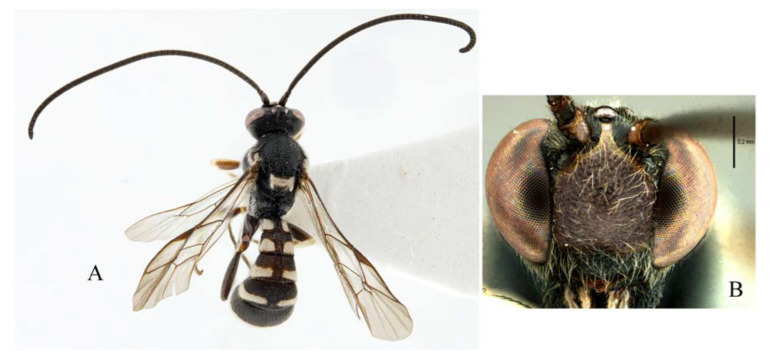
*Acerataspis separata* Liu, Reshchokv & Chen, sp. nov., holotype, female. (**A**) habitus, dorsal view; (**B**) head, frontal view. (Note: Facial colouration was altered after the DNA extraction experiment; the original colour of all species is yellow).

**Figure 25 insects-14-00389-f025:**
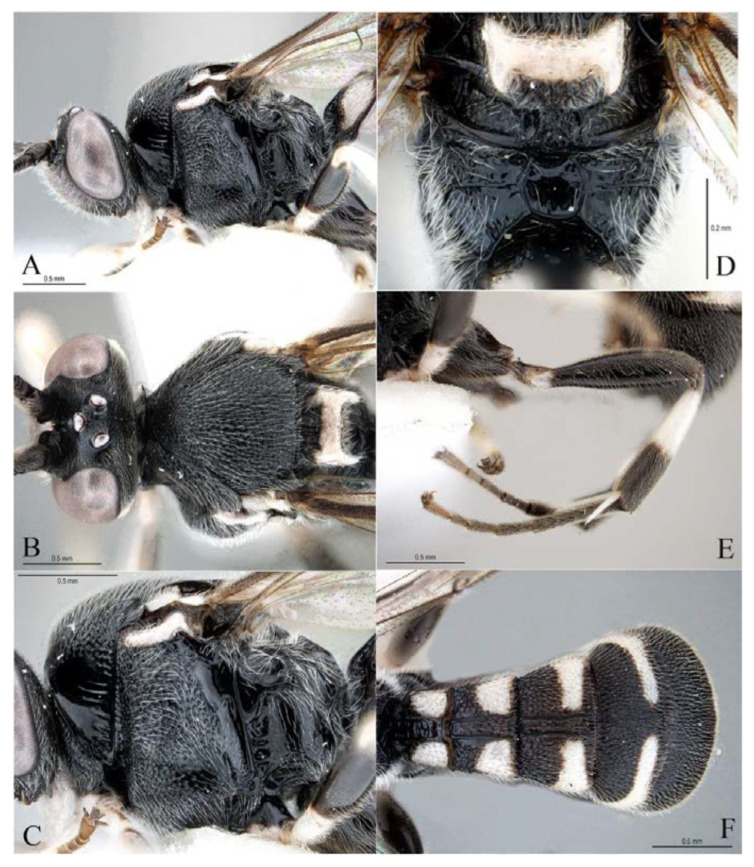
*Acerataspis separata* Liu, Reshchokv & Chen, sp. nov., holotype, female. (**A**) Head and mesosoma, lateral view; (**B**) head and mesosoma, dorsal view; (**C**) mesosoma, lateral view; (**D**) scutellum, metanotum and propodeum, dorsal view; (**E**) hind leg, lateral view; (**F**) metasoma, dorsal view.

**Figure 26 insects-14-00389-f026:**
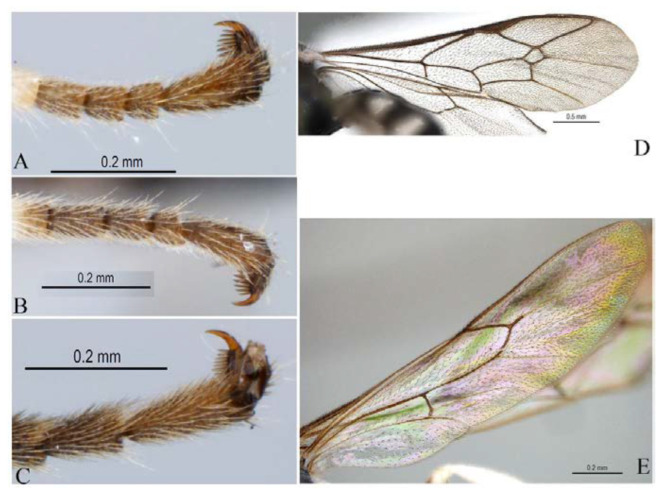
*Acerataspis separata* Liu, Reshchokv & Chen, sp. nov., holotype, female. (**A**) Fore tarsus; (**B**) middle tarsus; (**C**) hind tarsus; (**D**) wings; (**E**) hind wing.

**Figure 27 insects-14-00389-f027:**
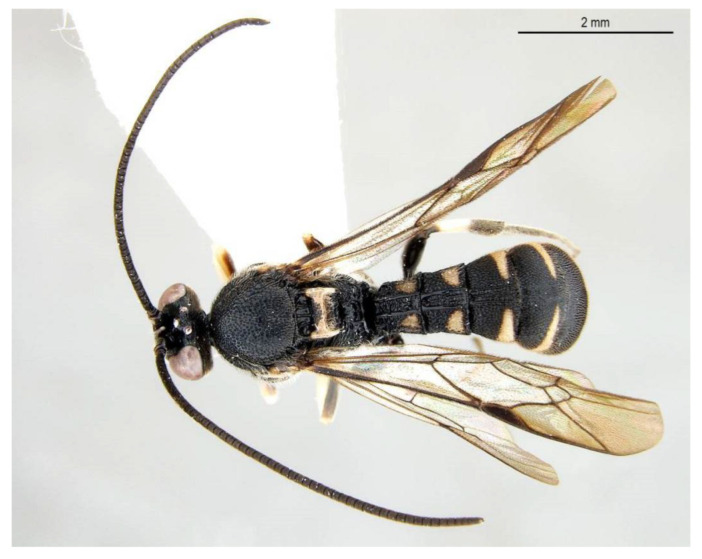
*Acerataspis separata* Liu, Reshchokv & Chen, sp. nov., paratype, male, habitus, dorsal view.

**Figure 28 insects-14-00389-f028:**
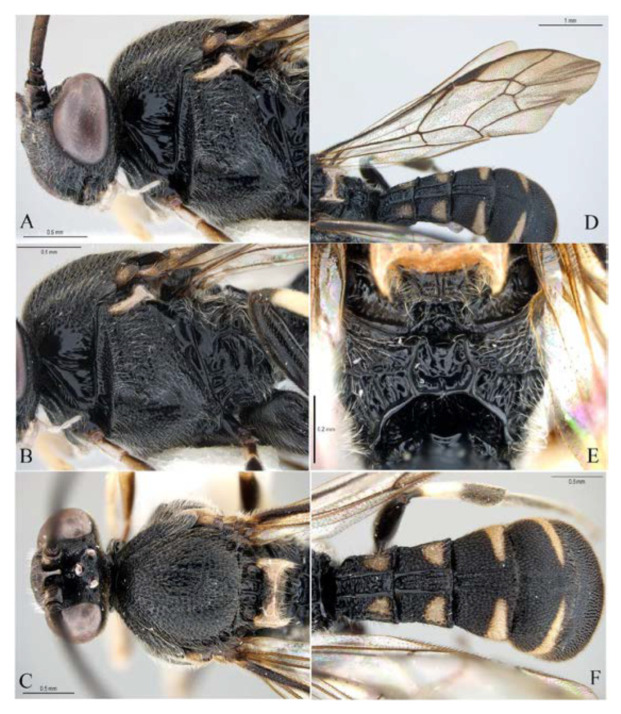
*Acerataspis separata* Liu, Reshchokv & Chen, sp. nov., paratype, male. (**A**) Head and anterior part of mesosoma, lateral view; (**B**) mesosoma, lateral view; (**C**) head and mesosoma, dorsal view; (**D**) wings; (**E**) posterior part of scutellum, metanotum and propodeum, dorsal view; (**F**) metasoma, dorsal view.

**Figure 29 insects-14-00389-f029:**
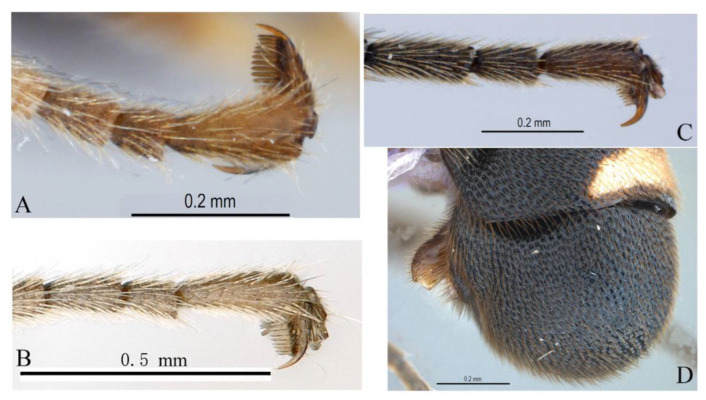
*Acerataspis separata* Liu, Reshchokv & Chen, sp. nov., paratype, male. (**A**) Fore tarsus; (**B**) middle tarsus; (**C**) hind tarsus; (**D**) T6 and paramere, lateral view.

#### 3.4.8. *Acerataspis similis* Liu, Reshchokv & Chen, sp. nov. (Figure 30, Figure 31 and Figure 32)

**Materials examined.** Holotype, 1♀, CHINA: Yunnan Province, Xishuangbanna, Menghai County, Bulangshan Town, grass, 21°44.521′ N, 100°26.647′ E, alt. 1646 m. 16.VIII–14.IX. 2018, Malaise trap, Li Ma leg., SCAU-3011981 (SCAU).

**Diagnosis.** This species can be distinguished from other species by the following combined characters: tarsal claws (Figure 32A–C) weakly and sparsely pectinate, fore tarsal claw with three pectinae, mid tarsal claw with four pectinae, hind tarsal claw with four pectinae (basal two slightly bigger than the apical two); fore wing with 1cu-a opposite M&RS (Figure 31D), areolet sessile and receiving 2m-cu before middle; metanotum black; hind leg black, hind tibia with a narrow whitish band (Figure 31E); T4 without carina (Figure 31F).

**Description.** Female, holotype (Figure 30A), body length 8.3 mm, fore wing length 6.3 mm.

Head. Face rugose punctate (Figure 30B), punctures close, some forming transverse wrinkles; moderately densely setose; combined face and clypeus 1.5 times as high as wide. Mandible slightly twisted, with upper margin longitudinally convex, upper tooth longer than the lower one. Malar space 0.5 times as long as the mandibular basal width. Head in dorsal view 2.2 times as wide as long. Temple very short, strongly narrowed behind eyes, 0.2 times as long as eye in dorsal view. POL:OD:OOL = 10:10:6. Vertex sloping vertically from posterior ocellus. Antenna with 58 flagellomeres, first flagellomere 1.1 times as long as its apical wide, 1.25 times as long as the second flagellomere.

Mesosoma. Pronotum (Figure 31A) with dorsal 0.3 strongly coarsely punctate and setose; lower 0.6 polished and glabrous, with several transverse carinae along the posterior margin (crenulate). Mesoscutum (Figure 31B) coarsely and closely punctate, lateral side of scutum with punctures closed and more or less forming wrinkles. Scuto-scutellar groove deeply crenulate, medially with three short longitudinal carinae. Scutellum transverse, punctate, lateral posterior corner with a sharp projection. Propodeum with area superomedia rectangular, 1.1 times as long as wide, anterior half weakly rugose, posterior half polished; anterior transverse carina joining area superomedia near middle; lateromedian longitudinal carinae subparallel; area externa with setiferous punctures, area dentipara laterally weakly rugose and setose; lateral longitudinal carina complete and strong; area spiracularis convered with long setae, pleural carina complete. Spiracle oval, connected to pleural carina. Mesopleuron (Figure 31C) strongly convex, densely and coarsely punctate, sternaulus broadly impressed. Epicnemial carina reaching to upper 0.7 of mesopleuron, with anterior end separated from the anterior margin of mesopleuron; mesopleuron posteriorly with a narrow longitudinal polished area along mesopleural furrow; mesoplerual furrow smooth, without fovea. Metapleuron (Figure 31C) convex, shiny, with fine setiferous punctures, anterior margin coarsely crenulate; juxtacoxal carina weak, interrupted on anterior 0.3. Submetapleural carina with an anterior flange.

Wings. Fore wing (Figure 31D) with 1cu-a, weakly inclivous, opposite to M&RS; 2r &RS slightly arched just behind the junction with pterostigma; 2rs-m:3rs-m = 13:11; areolet sessile and receiving 2m-cu before middle; 1/Cu:2cu-a = 16:7. Hind wing with nervellus interrupted at lower 0.3, distal abscissa of CU weakly pigmented.

Legs. Fore femur swollen, 1.88 times as long as its maximum width; fore tarsal claw sparsely pectinate, with three or four pectinae (Figure 32A), second to fifth tarsomeres ventrally flattened, shortly and densely pubescent. Middle femur 3.3 times as long as its maxium width, middle tarsal claw with four pectinae (Figure 32B). Hind femur (Figure 31E) 5.0 times as long as its maximum width, longest spur of hind tibia 0.63 times the length of basitarsus. Hind claw with four pectinae (Figure 32C), the basal two slightly longer and stronger than the apical two pectinae.

Metasoma. Strongly punctate. T1–T3 (Figure 31F) with a pair of latero-median carinae, area between latero-median carinae of T1 rugose punctate, that of T2 irregularly longitudinally wrinkled, that of T3 longitudinally rugose. Dorso-lateral carina of T1 absent behind spiracle. T1 as long as apical width, T2 0.88 times as long as apical width. T3 0.62 times as long as apical width, basal width:apical width = 8:13. T4 strongly punctate, transverse. Ovipositor sheath 0.73 times the length of hind basitarsus.

Colour. Body black with yellow maculation (Figure 30A). Face with upper 0.6 yellow (colour altered following DNA extraction), with lower 0.3 black. Mandible blackish brown with teeth apically lighter. Palpi yellowish brown. Antenna ventrally fuscous and dorsally blackish brown, scape yellow, laterally blackish, pedicel and annellus blackish. Tegula brown. Mesoscutum with posterior lateral corner yellow. Scutellum reddish brown. Subtegular ridge reddish. Wings hyaline, veins and pterostigma blackish brown. T1–T3 each with a pair of yellow spots subapically, the yellow maculation of T1 and T2 weakly connected medially, that of T3 interrupted dorso-medially (Figure 31F). T4 and T5 subapically with a yellow band. All coxae black (extreme apex of fore coxa brown); fore leg with trochanter dark brown, femur dark brown with apex yellowish brown, tibia and tarsus yellowish brown; middle leg with trochanter blackish brown, femur black, tibia blackish brown with basal 0.2 and apex yellowish brown; tarsus fuscous. Hind leg black, (except tibia subasally with a yellowish-brown band). Ovipositor sheath yellowish brown.

**Distribution.** Oriental region (China, Yunnan Province).

**Differential diagnosis.** This new species differs from *A. clavata* by fore wing with 1cu-a opposite to M&RS (postfurcal to M&RS of *A. clavata*), fore tarsal claw with 3–4 pectinae (with 8–10 pectinae in *A. clavata*). The new species is also quite similar to *A. fusiformis*, different in hind tarsal claw with four pectinae (hind tarsal claw with three pectinae in *A. fusiformis*) and black metanotum (metanotum yellow in *A. fusiformis*).

**Etymology.** The species name is derived from the Latin *similis*, referring to its colour pattern that is similar to *A. clavata* (Uchia, 1934).

**Figure 30 insects-14-00389-f030:**
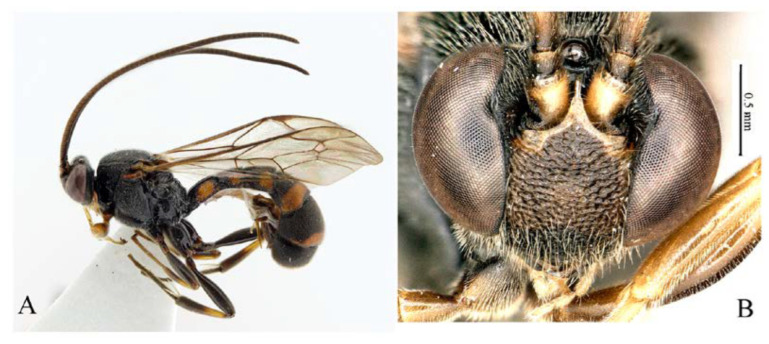
*Acerataspis similis* Liu, Reshchokv & Chen, sp. nov. holotype, female. (**A**) habitus lateral view; (**B**) head, frontal view. (Note: Facial colouration was altered after the DNA extraction experiment; the original colour of all species is yellow).

**Figure 31 insects-14-00389-f031:**
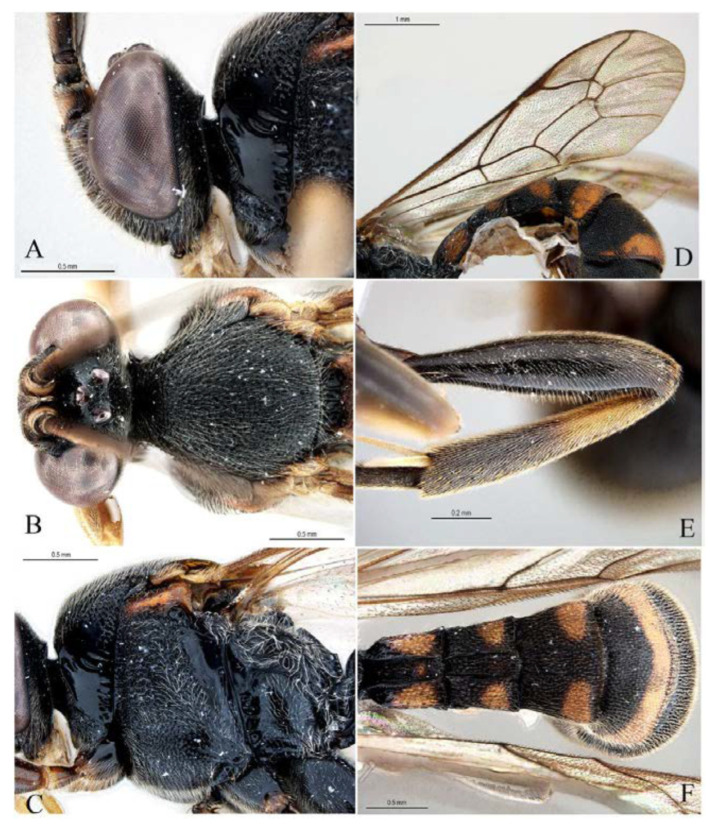
*Acerataspis similis* Liu Reshchokv & Chen, sp. nov., holotype, female. (**A**) Head and anterior part of mesosoma, lateral view; (**B**) head and mesosoma, dorsal view; (**C**) mesosoma, lateral view; (**D**) fore wing; (**E**) hind leg; (**F**) metasoma, dorsal view.

**Figure 32 insects-14-00389-f032:**
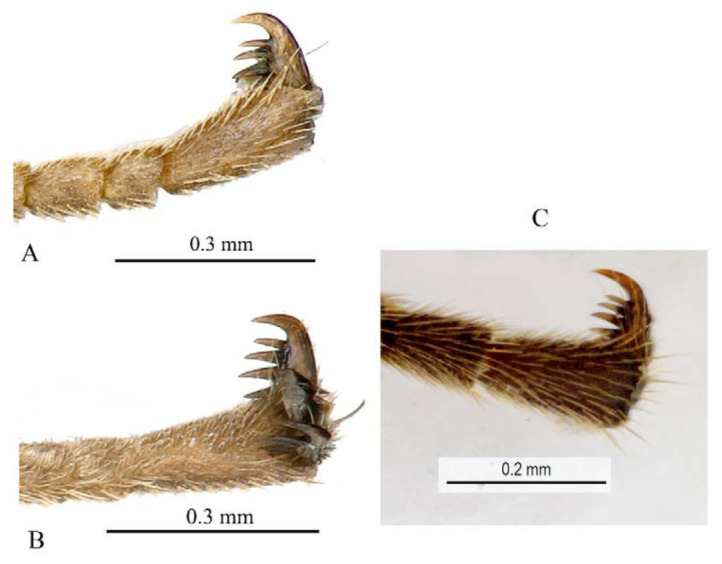
*Acerataspis similis* Liu Reshchokv & Chen, sp. nov., holotype, female. (**A**) Fore tarsal claw; (**B**) mid tarsal claw; (**C**) hind tarsal claw.

#### 3.4.9. *Acerataspis sinensis* Michener, 1940 (Figure 33 and Figure 34)

*Acerataspis sinensis* Michener, 1940, 123 [9]. Holotype from China (CAS).

**Materials examined.** Holotype, 1♀, labelled “Yim Na San E. Kwangtung, S. China VI-14–36; HOLOTYPE, *Acerataspis sinensis,* C. D. Michener” (CAS). **Other materials:** 1♀, CHINA: Zhejiang Province, West Mt. Tianmushan, 1982, Rui-liang Wang, No. 825,916 (ZJUH). labeled “*Acerataspis sinensis* Michener/He, 1983.XI.8”.

**Diagnosis.** Scuto-scutellar groove crenulate, with three median longitudinal carina, the central one strong. Scutellum strongly punctate (Figure 34A). Propodeum with the anterior transverse carina joining the area superomedia at apex. Fore wing with 1cu-a postfurcal to M&RS (Figure 34D); areolet slightly sessile (Figure 34B). Tarsal claws strongly pectinate, with 6–7 pectinae (Figure 34F). T1–T3 (Figure 34C) with a pair of median longitudinal carinae, area between carinae rugose punctate. T4 without longitudinal carina.

**Distribution.** Oriental and Palearctic regions [6].

**Figure 33 insects-14-00389-f033:**
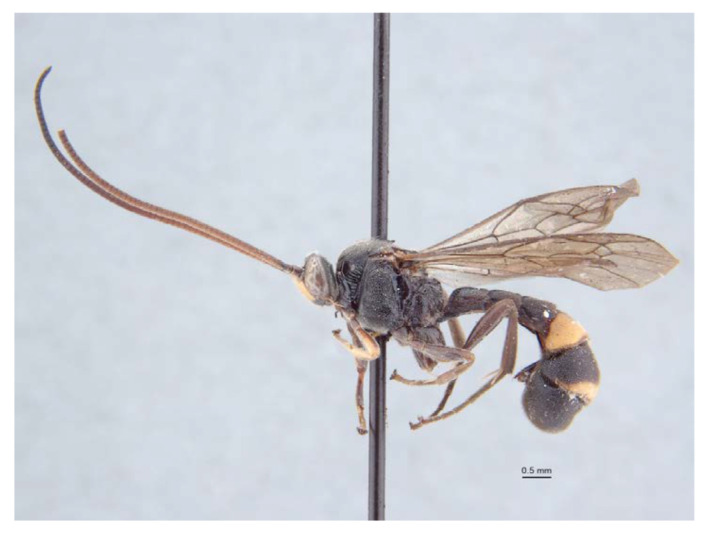
*Acerataspis sinensis* Michener, holotype, female, habitus, lateral view (Image: Robert Zuparko).

**Figure 34 insects-14-00389-f034:**
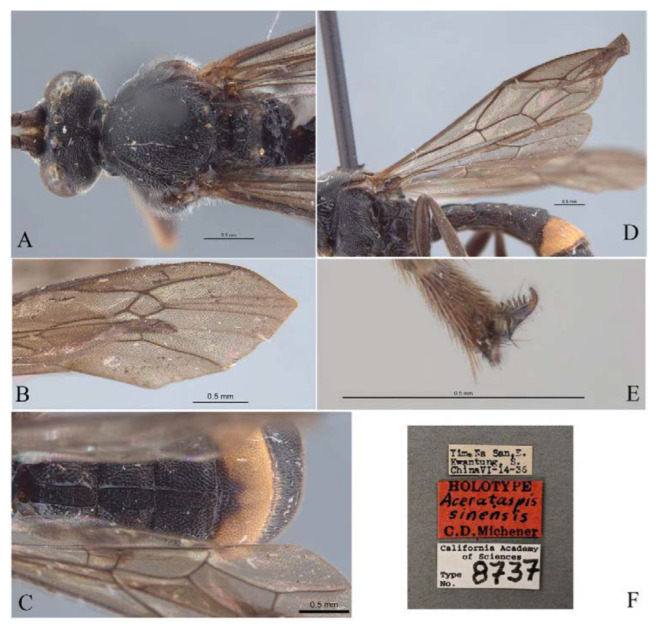
*Acerataspis sinensis* Michener, holotype, female. (**A**) Head and mesosoma, dorsal view; (**B**) apical part of fore wing; (**C**) metasoma, dorsal view; (**D**) wings; (**E**) hind tarsal claw; (**F**) labels (Images: Robert Zuparko).

#### 3.4.10. *Acerataspis szechuanensis* Chao, 1962 (Figure 35 and Figure 36)

*Acerataspis szechuanensis* Chao, 1962: 165 [3]. Holotype from China (IOZ).

**Materials examined.** Type materials, label (in Chinese) “Sichuan, E’mei Shan Qingyin Ge, 800–1000 m, 19857.IV.22; HOLOTYPE; *Acerataspis szechuanensis*, Chao ♀, Holotype” (IOZ).

**Diagnosis.** Propodeum with anterior transverse carina joining area superomedia at apex. Area between median longitudinal carina of T2 and T3 with an irregular weak carina (Figure 36A), T4 basally with a very short carina or wrinkle (Figure 36B). Fore tarsal claw with 6–7 pectinae (Figure 36C). Hind leg entirely blackish brown (Figure 35), hind tibia without white band basally.

**Distribution.** Oriental region (China, Sichuan Province).

**Differential diagnosis.** This species is very similar to *A. sinensis* in colour pattern; it is only different from *A. sinensis* in having scutellum black (yellow in *A. sinensis*) and recticulate punctation (normal punctuation in *A. sinensis*). It is probably a variant of *A. sinensis*; however, as there is no molecular data for either species, the status of *Acerataspis szechuanensis* is reserved.

**Figure 35 insects-14-00389-f035:**
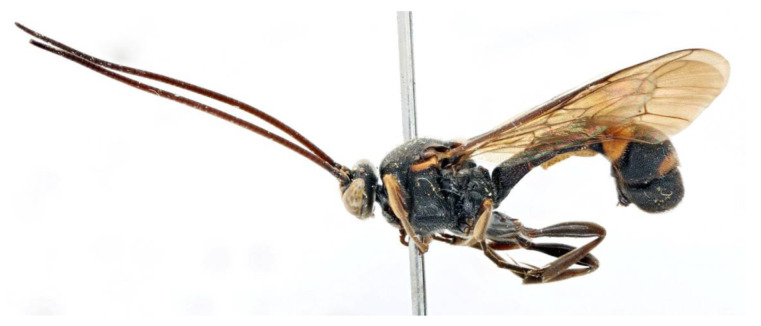
*Acerataspis szechuanensis* Chao, holotype, female, lateral view (Image: Huang Zhengzhong).

**Figure 36 insects-14-00389-f036:**
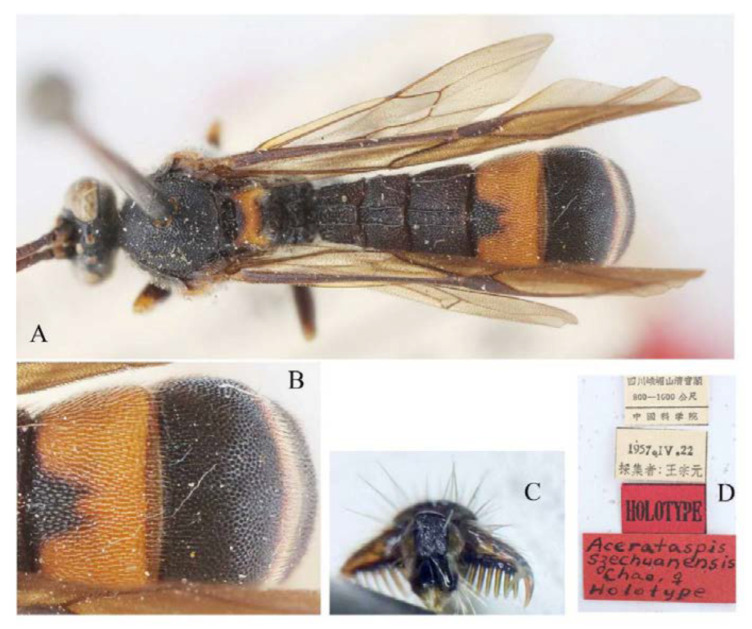
*Acerataspis szechuanensis* Chao, holotype, female. (**A**) Habitus, dorsal view; (**B**) T4 and T5, dorsal view; (**C**) fore tarsal claw, ventral view; (**D**) labels (Image: Huang Zhengzhong).

## 4. Discussion

Some studies have suggested that ichneumonid wasps have some characteristics— such as frequently occurring genetic introgression caused by *Wolbachia* endosymbionts and the low evolutionary rate of the *CO1*-marker—that make the relation between morphospecies and molecular species delimitation more complicated than in other Hymenoptera [33,34,35]. Therefore, the combination of morphological, molecular and some other characters have been proposed as a reliable method for species delimitation. In this study, we revised the species of *Acerataspis* using an integrative approach combining morphology and DNA barcoding techniques. In general, the morphological and molecular data complemented each other quite well in delimiting species. Compared with the *CO1* sequences of *A. clavata* newly generated in this study, the two sequences labeled as *A. clavata* downloaded from GenBank have an 8% basepair difference, suggesting that these two sequences may represent a different species or a haplotype given that these two sequences—together with the newly generated sequences—form a monophyletic clade (Figure 1). Similar situations where sequence data in public databases (e.g., Genbank and BOLD) might be based on misidentified specimens have been found in the genus *Enicospilus* (Ichnumonidae) [35].

In previous studies, the following characters were mainly used to separated different species in the key by various authors: the lateromedian longitudinal carinae of T1–T3 with or without an additional longitudinal carina [3,4,8,9,13,28]; the position where the anterior transverse carina of propodeum joins the area superomedian of propodeum; whether there is a pair of lateromedian longitudinal carinae on T4; the colour patterns of T1–T3; and the hind tibia is entirely black or black with a yellow band. However, besides the characters mentioned above, we found some additional morphological characters overlooked by previous studies that are also useful for the identification of *Acerataspis* species, especially in females: (1) the number of pectinae of claw pectination; (2) position between 1cu-a and M&RS of fore wing (opposite or postfurcal); (3) form of areolet in fore wing (sessile or petiolate); (4) apical yellow or white band of T4 complete or interrupted medially.

## Figures and Tables

**Figure 1 insects-14-00389-f001:**
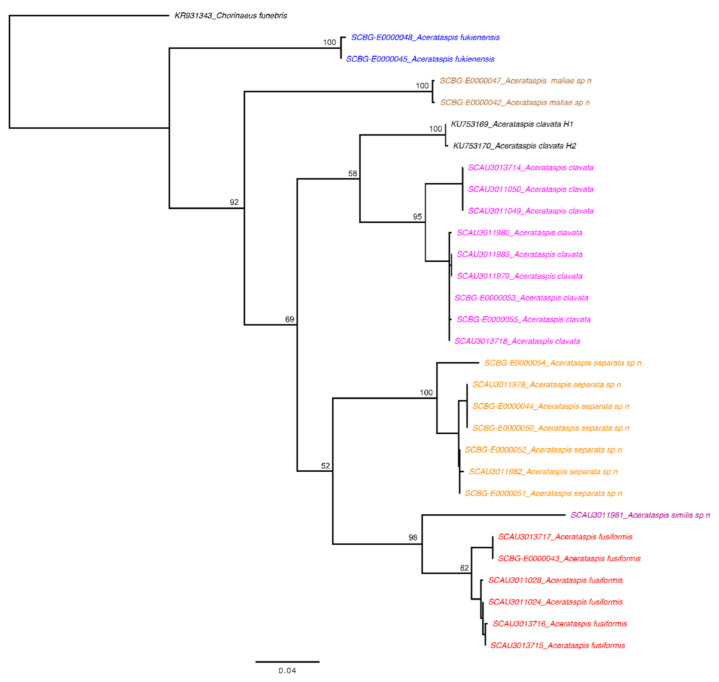
Maximum Likelihood tree demonstrating the clusters of *Acerataspis* species, based on *CO1*. The numbers at nodes show the boostrap values. The scale bar represents 0.04 substitutions per site.

**Table 1 insects-14-00389-t001:** List of sequenced taxa and accession numbers.

Code	Species	Sex	GanBank Accession No.
SCAU-3011979	*Acerataspis clavata* (Uchida, 1934)	female	OP525229
SCAU-3011980	*Acerataspis clavata* (Uchida, 1934)	male	OP525229
SCAU-3011983	*Acerataspis clavata* (Uchida, 1934)	male	OP525229
SCAU-3013718	*Acerataspis clavata* (Uchida, 1934)	female	OP525229
SCAU-3013714	*Acerataspis clavata* (Uchida, 1934)	male	OP525229
SCAU-3011049	*Acerataspis clavata* (Uchida, 1934)	male	OP525229
SCAU-3011050	*Acerataspis clavata* (Uchida, 1934)	female	OP525229
SCBG-E0000053	*Acerataspis clavata* (Uchida, 1934)	male	OP525229
SCBG-E0000055	*Acerataspis clavata* (Uchida, 1934)	male	OP525229
SCBG-E0000045	*Acerataspis fukienensis* Chao, 1957	male	OP525229
SCBG-E0000048	*Acerataspis fukienensis* Chao, 1957	female	OP525229
SCAU-3013715	*Acerataspis fusiformis* (Morley, 1913)	female	OP525229
SCAU-3013716	*Acerataspis fusiformis* (Morley, 1913)	male	OP525229
SCAU-3013717	*Acerataspis fusiformis* (Morley, 1913)	male	OP525229
SCAU-3011024	*Acerataspis fusiformis* (Morley, 1913)	male	OP525229
SCAU-3011028	*Acerataspis fusiformis* (Morley, 1913)	female	OP525229
SCBG-E0000043	*Acerataspis fusiformis* (Morley, 1913)	male	OP525229
SCBG-E0000042	*Acerataspis maliae* sp. n.	female	OP525229
SCBG-E0000047	*Acerataspis maliae* sp. n.	female	OP525229
SCAU-3011978	*Acerataspis separata* sp. n.	male	OP525229
SCAU-3011982	*Acerataspis separata* sp. n.	female	OP525229
SCBG-E0000044	*Acerataspis separata* sp. n.	male	OP525229
SCBG-E0000050	*Acerataspis separata* sp. n.	male	OP525229
SCBG-E0000051	*Acerataspis separata* sp. n.	male	OP525229
SCBG-E0000052	*Acerataspis separata* sp. n.	male	OP525229
SCBG-E0000054	*Acerataspis separata* sp. n.	male	OP525229
SCAU-3011981	*Acerataspis similis* sp. n.	female	OP525229

**Table 2 insects-14-00389-t002:** List of species of *Acerataspis* (Bold species indicate newly described species in this study).

Species	Distribution
*Acerataspis clavata* (Uchida, 1934)	Oriental and Palaearctic
*Acerataspis cruralis* Chiu, 1962	Oriental
*Acerataspis formosana* Cushman, 1937	Oriental
*Acerataspis fukienensis* Chao, 1957	Oriental
*Acerataspis fusiformis* (Morley, 1913)	Oriental
** *Acerataspis maliae* ** **sp. nov.**	Oriental
** *Acerataspis separata* ** **sp. nov.**	Oriental
** *Acerataspis similis* ** **sp. nov.**	Oriental
*Acerataspis sinensis* Michener, 1940	Oriental and Palaearctic
*Acerataspis szechuanensis* Chao, 1962	Oriental

## Data Availability

All data are available in this paper.

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
