# Peer review of "Descriptions of Three New Species of the Genus Acerataspis Uchida, 1934 (Hymenoptera, Ichneumonidae, Metopiinae), with an Illustrated Identification Key to Extant Species"

_insects, 2023, doi:10.3390/insects14040389_

Round 1

Reviewer 1 Report

This is a very useful partial revision of the genus Acerataspis.  This is a distinctive genus, mainly of South-east Asia, with lots of specimens in major collections, and this work is a significant improvement on any previous publications on Acerataspis.  The illustrations are excellent and the key works well, at least on specimens in NHMUK.  I was able to use the key to identify plenty of specimens of A. clavata from Nepal in NHMUK, and reidentify one of the specimens which had been misidentified as A. fusiformis.  Claw pectination is a great character, but difficult to see.  You don’t seem to mention in this paper that the apical flagellomere of males has a pale, weakly sclerotized area.  This is interesting in itself but is also a useful character for distinguishing species (it is nicely illustrated in, e.g., Figure 16bb and 20A).  For example, in NHMUK there is a series of specimens from Brunei and Malaysia which are close to A. clavata but males clearly differ in the apical flagellomere, which has an extensive pale/weakly sclerotized area.  This brings me to the only real weakness of the paper, which is that it is not really a taxonomic revision in the true sense of a ‘revision’.  It is a paper with descriptions of three new species from a few localities, with a key to described species.  A revision would involve studying the available material.  Additional undescribed species are known from collections and from additional countries, such as from the Philippines, in the Townes collection, as you mention.  In NHMUK, I think there are four undescribed species (plus one potential specimen of A. separata, from Brunei).  There are more known and available species of Acerataspis in collections, so not including these means it is not really a revision, although it is a very useful work.

The two species, A. similis and A. fusiformis, are very similar, and as far as I can see are separated on the basis of whether there are three or four teeth (or pectinae) on the hind claw, and whether the metanotum is yellow or black (although you don’t include the metanotum colour in the description of A. similis).  These are not significant differences.  I presume you were prompted to describe a new species based on the significant barcoding gap, but you don’t state that.  Most of the discussion centres on the separation of A. fusiformis and A. clavata, but these two species seem more obviously distinct.

I have made some minor corrections and suggestions on the PDF.  It is worth noting that a few A. clavata females have vein 1cu-a opposite M&RS.  Check all the morphological terminology again; there are some inconsistencies such as metapleuron vs metascutellum vs metascutum, areas of the propodeum sometimes misspelt or hyphenated.  The correct term for the individual teeth on the tarsal claws is ‘pectinae’.  The ‘pecten’ is the whole comb of teeth.

You don’t explain why full descriptions are given for most species but not for A. sinensis nor A. szechuanensis.

Author Response

Many thanks for your constructive review! 

In the text, all corrections were highlighted in blue (review 1) or red (reviewer 2). Also, we have placed all the images as suggested.

Our replies to your comments or questions are as follows:

Q1: Claw pectination is a great character, but difficult to see.

Reply: We agree that claw pectination is difficult to see. However, it is easy to observe the pectination when the magnification of a stereomicroscope is above 40×.

Q2: You don’t seem to mention in this paper that the apical flagellomere of males has a pale, weakly sclerotized area. This is interesting in itself but is also a useful character for distinguishing species (it is nicely illustrated in, e.g., Figure 16bb and 20A).

Reply: We agree that the apical flagellomere of males has a pale and weakly sclerotized area in some species, especially the differences of it between A. clavata and A. fusiformis (in the key to species, we use the ratio of length to width). We didn’t mention the character in the descriptions, for following reasons: 1) only five species both with female and male were known, we didn’t sure whether this character is common in males of the other known species; 2) the character is very useful for distinguishing A. clavata and A. fusiformis from other species, however, in A. fukienensis and A. separata, the apical flagellomere is not so specialized as that in A. clavata and A. fusiformis. Nevertheless, we agree that it is useful to distinguish species by combining other characters as well. We have added this character in the diagnosis section in A. clavata and A. fusiformis.

Q3: This brings me to the only real weakness of the paper, which is that it is not really a taxonomic revision in the true sense of a ‘revision’. It is a paper with descriptions of three new species from a few localities, with a key to described species.  A revision would involve studying the available material. Additional undescribed species are known from collections and from additional countries, such as from the Philippines, in the Townes collection, as you mention.  In NHMUK, I think there are four undescribed species (plus one potential specimen of A. separata, from Brunei).  There are more known and available species of Acerataspis in collections, so not including these means it is not really a revision, although it is a very useful work

Reply: We agree that the specimens sources of our paper are narrowly focus on a few localities, due to a few obstacles that prevented us travel around the world to check other material of Acerataspis from different museums during the COVID pandemic. We now use a more  appropriate title “A study on the genus Acerataspis (Hymenoptera, Ichneumonidae, Metopiinae), with descriptions of three new species”.

Q4: The two species, A. similis and A. fusiformis, are very similar, and as far as I can see are separated on the basis of whether there are three or four teeth (or pectinae) on the hind claw, and whether the metanotum is yellow or black (although you don’t include the metanotum colour in the description of A. similis).  These are not significant differences.  I presume you were prompted to describe a new species based on the significant barcoding gap, but you don’t state that.

Reply: We have added some descriptions in the text. The new species A. similis is based on the significant barcoding gap as well as recognizable morphological chatacters, we have added some explanation and comments in the text.

Q5: Most of the discussion centres on the separation of A. fusiformis and A. clavata, but these two species seem more obviously distinct

Reply: We have moved the discussion on A. fusiformis and A. clavata to the “comments” section of A. fusiformis. The colour pattern of these two species are very similar to each other, in some collections many specimens of A. fusiformiswere mistakenly determined as A. clavata, in the case, we think it is necessary to focus on these two species in the “discussion’ section. Base on the comments of both reviewers, we agree to move this discussion to the “comments” sections of A. fusiformis.

Q6: I have made some minor corrections and suggestions on the PDF.  It is worth noting that a few A. clavata females have vein 1cu-a opposite M&RS.  Check all the morphological terminology again; there are some inconsistencies such as metapleuron vs metascutellum vs metascutum, areas of the propodeum sometimes misspelt or hyphenated.  The correct term for the individual teeth on the tarsal claws is ‘pectinae’.  The ‘pecten’ is the whole comb of teeth.

Reply: We have made the corrections in the text base on the comments and suggestions. For the characters of A clavata, we have added some comments on variations. We have checked all the morphological terminology and corrected some misspelt vocabulary, some hyphenated was caused by the text wrap around. The “pecten” was replaced by “pectinae”.

Q7:  You don’t explain why full descriptions are given for most species but not for A. sinensis nor A. szechuanensis.

Reply: Full descriptions are given for three new species, A. fukienensis and A. fusiformis. In the original description, A. fukienensis was described in Chinese and with a simple English summary, and A. fusiformis was briefly described, therefore, we redescribed these two species. The other previously described species including A. sinensis and A. szechuanensis were well described, in the original paper. Since we don’t have new information to add to the description, we decided not to provide a re-description.  

Replies to the comments in the text by reviewer 1

Q1 in Line 36: Why would Townes provide label data? The point was to make the information available, that there are /were undescribed species from Indonesia and the Philippines.

Reply: We have rewritten this part.

Q2 in line 158 I wouldn't describe the ovipositor as 'needle-like'. It doesn't have a distinct notch or teeth but it isn't as thin as, for example, a mesochorine ovipositor, and it has a shallow concavity on the upper valve.

Reply: We agree with the opinion of the reviewer and have rewritten the sentence in the text.

Q3 in line 437: This is not a species-level character: females always have two spurs.

Reply: Females have two mid tibial spurs, but the description is about males, male of Acerataspis usually with a single mid tibial spur, but our study have the male of A. fukienensis with two mid tibial spurs. So, we think it is necessary to point it out.

Q4 in line 460 : I don't understand this bit. Why is the description not based on the specimens you examined? Do you mean that you were unable to examine the holotype? From the photos of the holotype which you include here, many features can be seen besides the wings and legs.

Reply: Yes, we have examined the holotype of Chao. The sentence was written in the early draft of the manuscript. We have deleted the sentence in the text.

Q5 in line 806 If the colour of the metanotum was the only difference, I don't think that would be a good indicator of it being a different species. The colour varies in some species. In the key you also use the number of pectinae to separate the two species.

Reply: This species is very similar to A. fusiformis, besides the colour of the metanotum, the number of pectinae is a good character to separate the two species. We have added the number of pectinae to separate the two species as well.

Reviewer 2 Report

This is a very nice manuscript except for format, terms, and discussion. There are many unnecessary blank lines and the titles of the sections are sometimes not bolded. Regarding morphological terminologies, you said that you follow those of Broad et al. (2018), however, there are many errors. I've indicated errors in formats and terms as well as possible, but there are too many errors (I've added 666 comments or changes in the attached PDF) and probably I missed some errors, so please carefully check your manuscript. With respect to the discussion, some parts are mentioned about the very narrow focus of the subject and not enough discussed, indicating that such part should be moved to the "Comments" section in each species part. I strongly recommend you reconstruct the "Discussion" section logically.

While there are too many minor mistakes, this paper is taxonomically very high quality and I think it is worth publishing. I'm really looking forward to publishing this manuscript.

Author Response

Thanks for the constructive review! 

In the text, all corrections were highlighted in blue (review 1) or red (reviewer 2). Also, we have placed all the images as suggested.

Our replies to your comments/questions are as follows:

Q1: There are many unnecessary blank lines and the titles of the sections are sometimes not bolded.

Reply: We have modified the manuscript base on the comments in the text, and have checked carefully throughout the text, all corrections were highlighted in red.

Q2: With respect to the discussion, some parts are mentioned about the very narrow focus of the subject and not enough discussed, indicating that such part should be moved to the "Comments" section in each species part. I strongly recommend you reconstruct the "Discussion" section logically.

Reply: We agree with the reviewer’s comment and have revised the “discussion” section. The paragraph on discussion between “A. clavata” and “A. fusiformis” was moved to the “comments” section of “A. fusiformis”.

Replies to the comments of Reviewer 2 in the text

Line 17-19 Results should be written in the past tense, and discussion and conclusion should be in the present tense. This sentence can be included in the first sentence of abstract.

Reply: We agree and we have used past tense.

Line 46-51 I think that this part is NOT introduction and should be moved to result section. In introduction section, you should not mention about result and just describe about what will you do in this paper. For example, you can write as follows: "In this study, we examined the species of Acerataspis from Asia and revised it based on morphological and molecular analyses, with an illustrated key to species."

Reply: We have reconstructed the paragraph in the text, and moved to “result” section.

Line 106-113 Why you would like to reconstruct phylogenetic tree?

-> If you would like to reveal phylogenetic affinities, you should mention about it.

-> If you would like to delimit molecular operational taxonomic units (MOTUs), you should mention about it and run species delimitation software, such as PTP and ABGD (you can very easily to run this program using aligned sequences or tree file with branch length).

Reply: By constructing a phylogenetic tree, we would like to show the phylogentic affinities among the studied species.

Line 120-122 You should add section about the result of morphological species identification. How many specimens were examined? How many morphospecies were recognized? How many species were previously described species among the morphospecies? etc.

Reply: We have added these results.

Line 134 The tree should not be compressed, because intraspecific divergence could not be understand from the compressed leaf. Additionally, I'd like to know how did you ultrametricized this tree, it is not explained in the M&M section.

Reply: we have provided a new figure for the tree. And some explanations were added in the M& M section in red.

Line 159 This section is redundant to explain importance of metopiine tarsal claw pectinations.

Reply: we agree with the reviewer’s comment and rewrote the paragraph.

Line 282; How many individuals and which sexes?

Reply: we have added the information in the text.

Line 296. This is apparently not variation and should be moved to comments or you should create "DNA barcodes" section

Reply: we have revised this paragraph.

Line 517 When I read this section, materials from some different localities were included in "female", "variation", and "male" description. This absolutely make me (and readers) confusing, and you must improve these issues.

Reply: we have revised this paragraph base on the suggestions.

Line 766-771 I think this part can describe more clearly. For example: "This new species and A.clavata from Thailand (SCAU-3013718) share the separated maculations of T4 and possibly confused, but are distinguished by the dorsal color of scape (black in the new species vs. yellow in Thailand A. clavata) and number of metapleural foveae  (three in the new species vs. two in Thailand A. clavata)."

Reply: We agree with the reviewer’s comments and rewrote this paragraph base on the suggestions.

Line 933-936 The authors discussed only about A. fusiformis and A. clavata here, so I'd like to recommend the following changes:

(1) change "3. Result" section to "3. Result and discussion"

(2) move discussion about A. fusiformis and A. clavata to "comment" section of fusiformis or clavata.

(3) delete "4. discussion" section

If you would like to keep this section, you should discuss about more general subjects.

Reply: We agree with the reviewer’s comments, the discussion about “A. clavata” and A. fusiformis was moved to the “comment” section of “A. fusiformis”. The “discussion” was reconstructed base on the (2) suggestion.

Round 2

Reviewer 2 Report

The manuscript has been much improved. However, some needed corrections are found. All comments are indicated in the attached PDF file. Please find it.

Author Response

Hello,

We have revised the manuscript according to the comments of the reviewer. All the changes we have made can be easily found as we use the “Track Changes” function. Since there are many minor changes that we were suggested to make, here we include our replies to a few major points.

  1. We have revised the descriptions about the method related to the molecular analyses. And accordingly, we have replaced the image of the phylogenetic tree.
  2. We have revised the discussion on the DNA barcoding results.
  3. We have revised the discussion on the morphological characters that are useful for the identification of Acerataspis.

We hope that the revision we have made would meet the satisfaction. 
